# Taf2 mediates DNA binding of Taf14

Brianna J. Klein[1], Jordan T. Feigerle[2,3], Jibo Zhang[4], Christopher C. Ebmeier [5], Lixin Fan[6], Rohit K. Singh[1], Wesley W. Wang[7], Lauren R. Schmitt[8], Thomas Lee[6], Kirk C. Hansen[8], Wenshe R. Liu [7], Yun-Xing Wang [9], Brian D. Strahl [4], P. Anthony Weil[3] & Tatiana G. Kutateladze [1✉]

The assembly and function of the yeast general transcription factor TFIID complex requires specific contacts between its Taf14 and Taf2 subunits, however, the mechanism underlying these contacts remains unclear. Here, we determined the molecular and structural basis by which the YEATS and ET domains of Taf14 bind to the C-terminal tail of Taf2 and identified a unique DNA-binding activity of the linker region connecting the two domains. We show that in the absence of ligands the linker region of Taf14 is occluded by the surrounding domains, and therefore the DNA binding function of Taf14 is autoinhibited. Binding of Taf2 promotes a conformational rearrangement in Taf14, resulting in a release of the linker for the engagement with DNA and the nucleosome. Genetic in vivo data indicate that the association of Taf14 with both Taf2 and DNA is essential for transcriptional regulation. Our findings provide a basis for deciphering the role of individual TFIID subunits in mediating gene transcription.

[1] Department of Pharmacology, University of Colorado School of Medicine, Aurora, CO 80045, USA. [2] Department of Structural Biology, Stanford University, Stanford, CA 94305, USA. [3] Department of Molecular Physiology and Biophysics, Vanderbilt University School of Medicine, Nashville, TN 37232, USA. [4] Department of Biochemistry & Biophysics, The University of North Carolina School of Medicine, Chapel Hill, NC 27599, USA. [5] Department of Biochemistry, University of Colorado, Boulder, CO 80303, USA. [6] Basic Science Program, Frederick National Laboratory for Cancer Research, SAXS Core Facility of the National Cancer Institute, Frederick, MD 21702, USA. [7] Department of Chemistry, Texas A&M University, College Station, TX 77843, USA. [8] Department of Biochemistry and Molecular Genetics, University of Colorado School of Medicine, Aurora, CO 80045, USA. [9] Protein-Nucleic Acid Interaction Section, Structural Biophysics Laboratory, Center for Cancer Research, National Cancer Institute, National Institutes of Health, Frederick, MD 27102, USA. ✉email: tatiana.kutateladze@cuanschutz.edu

The initiation of gene transcription by RNA polymerase II (RNA Pol II) requires formation of the preinitiation complex (PIC) around the gene promoter region. In addition to RNA Pol II and a coactivator, also known as Mediator, the PIC contains six general transcription factor (TF) complexes, including TFIID and TFIIF. TFIIF is involved in the recruitment of RNA Pol II and the association with Mediator, whereas TFIID is responsible for binding to the promoter. Both yeast TFIID and TFIIF contain multiple subunits and share one subunit – Taf14 (transcription initiation factor subunit 14)[1–5]. Additionally, Taf14 has been implicated in chromatin-remodeling and histone-modifying processes, being a component of the INO80, SWI/SNF and RSC complexes[3,6,7] and the histone acetyltransferase complex NuA3[8], and therefore plays numerous vital roles in chromatin biology. Taf14 is a 244-residue protein, which consists of the N-terminal YEATS (Yaf9, ENL, AF9, Taf14, Sas5) domain and the extra-terminal (ET) domain (Fig. 1a). The YEATS domain of Taf14 recognizes acylated lysine 9 of histone H3 (H3K9acyl)[9–12], and the ET domain is capable of associating with the hxhKh motif found in several co-activators, such as Sth1 and Taf2[13]. Genetic studies have shown that the interaction between Taf14 and another subunit of TFIID complex, Taf2, could be essential in the TFIID assembly and function[14], however the mechanism underlying this interaction remains unclear.

In this study, we describe the molecular mechanism by which the YEATS and ET domains of Taf14 bind to the C-terminal tail of Taf2 and characterize a DNA-binding activity of the linker connecting the two domains. We show that Taf14 undergoes a conformational rearrangement upon binding to Taf2 to release the linker for binding to DNA – a function that is important for the ability of Taf14 to regulate gene transcription.

## Results and discussion

**Structural insight into the Taf14-Taf2 complex formation.** To define the molecular basis for the interaction between Taf14 and Taf2, we generated a set of truncated constructs of Taf14 and examined their association with the Taf2 peptide corresponding to the C-terminal hxhKh motif-containing tail of Taf2 (Taf2$_{CT}$) by NMR and microscale thermophoresis (MST). Titration of Taf2$_{CT}$ to the $^{15}$N-labeled ET domain of Taf14 (Taf14$_{ET}$) led to large chemical shift perturbations (CSPs) in the $^{1}$H$^{15}$,N heteronuclear single quantum coherence (HSQC) NMR spectra of Taf14$_{ET}$, indicating formation of the complex (Fig. 1b). Many amide crosspeaks of Taf14$_{ET}$ in the apo-state disappeared upon addition of the Taf2$_{CT}$ peptide, and another set of resonances corresponding to the Taf2$_{CT}$-bound state of Taf14$_{ET}$ appeared. The slow exchange regime on the NMR timescale suggested tight binding (Fig. 1c), which was confirmed by measuring binding affinity of Taf14$_{ET}$ for Taf2$_{CT}$ using MST (the dissociation constant ($K_d$) = 1.6 μM) (Fig. 1c).

To gain insight into the binding mechanism, we crystallized the Taf14$_{ET}$-Taf2$_{CT}$ complex and determined its structure to 1.7 Å by X-ray crystallography (Fig. 1d–f and Supplementary Table 1). Taf14$_{ET}$ adopts a three-helix bundle fold with a two-stranded anti-parallel β-sheet connecting helices α2 and α3. The Taf2$_{CT}$ peptide is in an extended conformation and occupies an elongated, primarily hydrophobic groove of Taf14$_{ET}$ (Fig. 1d). Taf2$_{CT}$ pairs with the second β-strand of Taf14$_{ET}$, forming the third anti-parallel β-strand in the β-sheet, and makes extensive intermolecular contacts with the domain (Fig. 1e). Characteristic β-sheet interactions are observed between the backbone amides of V1397, I1399 and T1401 of Taf2$_{CT}$ and G218, F220 and I222 of Taf14$_{ET}$. The side chains of V1397 and I1399 of Taf2$_{CT}$ are oriented toward and essentially buried in the hydrophobic groove

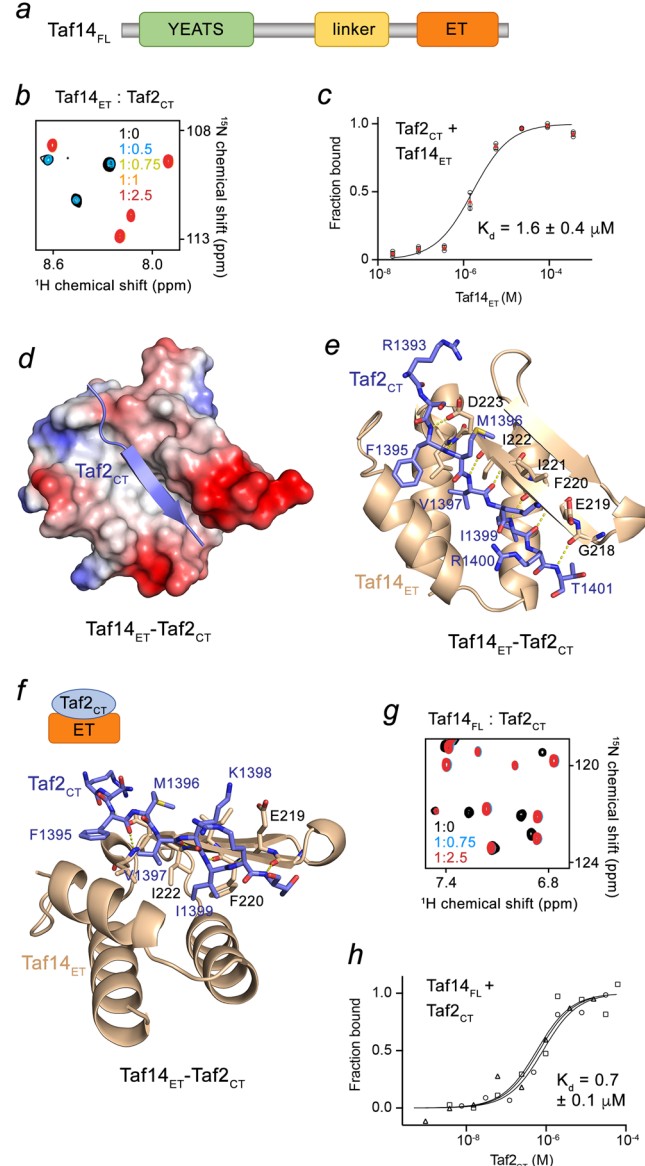

**Fig. 1 Structural mechanism for the interaction of Taf14$_{ET}$ with Taf2$_{CT}$. a** Architecture of Taf14$_{FL}$. **b** Superimposed $^{1}$H$^{15}$,N HSQC spectra of Taf14$_{ET}$ recorded in the presence of increasing amounts of the Taf2$_{CT}$ peptide. The spectra are color coded according to the protein:peptide molar ratio. **c** MST binding curve for the FAM-Taf2$_{CT}$ interaction with Taf14$_{ET}$. The $K_d$ value represents average (shown as red diamonds) of three independent measurements (open circles), and error bars represent the standard deviation from average. $n = 3$ **d** Electrostatic surface potential of the crystal structure of the Taf14$_{ET}$:Taf2$_{CT}$ complex, with blue and red colors representing positive and negative charges, respectively. The Taf2$_{CT}$ peptide is shown as a purple ribbon. **e** Ribbon diagram of the Taf14$_{ET}$:Taf2$_{CT}$ complex structure in the same orientation as in (**d**). Taf14$_{ET}$ domain is colored wheat and Taf2$_{CT}$ peptide is purple. Residues involved in the interaction between Taf2$_{CT}$ and Taf14$_{ET}$ are labeled. Dashed lines represent hydrogen bonds. **f** Rotated view of the ribbon structure shown in (**e**). Schematic of the Taf14$_{ET}$-Taf2$_{CT}$ complex is shown above the structure. **g** Superimposed $^{1}$H$^{15}$,N HSQC spectra of Taf14$_{FL}$ recorded in the presence of increasing amounts of the Taf2$_{CT}$ peptide. The spectra are color coded according to the protein:peptide molar ratio. **h** MST binding curves for the FAM-Taf2$_{CT}$ interaction with Taf14$_{FL}$. The $K_d$ value represents average of three independent measurements, and error represents the standard deviation. $n = 3$.

of Taf14$_{ET}$. The complex is further stabilized by a set of additional hydrogen bonds formed between the backbone amide group of F1395 of Taf2$_{CT}$ and the side chain carboxyl group and the backbone amino group of D223 of Taf14$_{ET}$. The side chain amino group of K1398 of Taf2$_{CT}$ donates a transient hydrogen bond to the side chain carboxyl moiety of E219 of Taf14$_{ET}$, whereas the guanidino moiety of R1400 of Taf2$_{CT}$ is hydrogen bonded to the backbone carbonyl group of K1398.

To explore whether the interaction with Taf2$_{CT}$ is conserved in full-length Taf14 (Taf14$_{FL}$), we expressed Taf14$_{FL}$ as an $^{15}$N-labeled protein and tested it in NMR titration experiments. The pattern of CSPs induced in Taf14$_{FL}$ by Taf2$_{CT}$ and fast saturation suggested a stronger interaction compared to the binding of Taf14$_{ET}$ (Fig. 1g). In agreement, the K$_d$ value for the interaction of Taf14$_{FL}$ with Taf2$_{CT}$ was found to be 0.7 μM (Fig. 1h), demonstrating that binding of the full-length protein is ~two-fold tighter than binding of the isolated Taf14$_{ET}$.

**Both domains of Taf14, Taf14$_{ET}$ and Taf14$_{YEATS}$, bind to Taf2.** In effort to explain the increase in binding affinity of Taf14$_{FL}$ toward Taf2$_{CT}$, we examined if other parts of Taf14 can contribute to the interaction with Taf2. Surprisingly, we found that the YEATS domain of Taf14 (Taf14$_{YEATS}$) is capable of binding to Taf2$_{CT}$ as titration of the Taf2$_{CT}$ peptide into the $^{15}$N-labeled Taf14$_{YEATS}$ caused substantial CSPs in NMR spectra of Taf14$_{YEATS}$ (Fig. 2a). MST measurements revealed that Taf14$_{YEATS}$ interacts with Taf2$_{CT}$ with a K$_d$ of 6 μM (Fig. 2b). Taf14$_{YEATS}$ is a well-established reader of H3K9acyl, particularly the crotonyl modification, H3K9cr[9,11]. Titration of the H3K9cr peptide into the Taf14$_{YEATS}$ NMR sample resulted in CSPs that were distinctly different from CSPs caused by titration of Taf2$_{CT}$, implying that Taf14$_{YEATS}$ has separate binding sites for the two ligands, H3K9cr and Taf2$_{CT}$ (Fig. 2c, d and Supplementary Fig. 1). Indeed, the structure of the Taf14$_{YEATS}$-H3K9cr complex and the mapping of the most perturbed residues in NMR titration experiments showed that the H3K9cr-biding site of Taf14$_{YEATS}$ is located at one of the open ends of the Taf14$_{YEATS}$ β-sandwich (Fig. 2e, orange), whereas residues located at the opposite side of the β-sandwich, in particular S89, G97, E98, K100 and I101, were perturbed upon binding of Taf2$_{CT}$ (Fig. 2f, magenta).

Evaluation of the TFIIF complex, and particularly Taf14$_{FL}$, in the previously reported cryo-EM structure of the PIC[4] showed that Taf14$_{YEATS}$ and Taf14$_{ET}$ are in close proximity (Fig. 2g). An overlay of the cryo-EM structure of Taf14$_{FL}$ (grey) with the structure of the Taf14$_{ET}$(wheat)-Taf2$_{CT}$(blue) complex and the Taf14$_{YEATS}$(green)-H3K9cr complex (with the Taf2$_{CT}$-binding site colored magenta) suggests that Taf2$_{CT}$ could be concurrently engaged with both domains, Taf14$_{ET}$ and Taf14$_{YEATS}$ (Fig. 2g and Supplementary Fig. 2). The possibility of such an engagement was corroborated by mass spectrometry analysis of DSSO cross-linked Taf14$_{FL}$ in the presence of Taf2$_{CT}$ (Fig. 2h, Supplementary Fig. 3 and Supplementary Data 1). In the Taf14$_{ET}$-Taf2$_{CT}$ complex, the only lysine residue of Taf2$_{CT}$, K1398, is in the middle of the β-strand and thus its position is fixed. The side chain of K1398 is perpendicular to the Taf14$_{ET}$ surface, pointing straight away from the domain (Supplementary Fig. 3), and cannot be easily crosslinked backward to Taf14$_{ET}$, as only a single weak contact was detected by MS (Fig. 2h). However, K1398 is readily crosslinked with several lysine residues of Taf14$_{YEATS}$, located in the same part of Taf14$_{YEATS}$ that was perturbed in NMR experiments (Fig. 2h and Supplementary Fig. 3). The hydrophobic residues V1397 and I1399 of Taf2$_{CT}$ were found to be required for this engagement, as V1397D and I1399D mutations in Taf2$_{CT}$ substantially reduced binding of Taf14$_{ET}$ and abrogated binding of Taf14$_{YEATS}$ (Fig. 2i).

**Taf14$_{ET}$ and Taf14$_{YEATS}$ regulate DNA binding activity of the linker (Taf14$_{Linker}$).** Because TFIID plays an essential role in binding to DNA and bridging the PIC to gene promoters, we tested whether Taf14 has a DNA binding capability by NMR and electrophoretic mobility shift assays (EMSA). The direct interaction of Taf14$_{FL}$ with 601 Widom DNA was initially detected by CSPs in $^1$H$^{15}$N HSQC spectra of Taf14$_{FL}$ (Fig. 3a) and then confirmed in EMSA experiments (Fig. 3b). In EMSA, 601 DNA was incubated with increasing amounts of Taf14$_{FL}$, and the reaction mixtures were resolved on a native polyacrylamide gel. A shift of the free 601 DNA indicated the formation of Taf14$_{FL}$:DNA complex (Fig. 3b). To identify the DNA binding region, we generated a set of truncated and mutated Taf14 constructs and tested them in EMSA (Fig. 3c–e and Supplementary Fig. 4a–d). We found that Taf14$_{YEATS}$ does not bind to 601 DNA, supporting previous findings[15], and Taf14$_{ET}$ shows a very weak DNA binding activity (Supplementary Fig. 4a, b). The W81A mutation in Taf14$_{FL}$ that impairs binding of Taf14$_{YEATS}$ to H3K9cr had no effect on the DNA binding (Supplementary Fig. 4c), whereas separate K149E/R150E/R151E (3K/Rmut) and K161E/K163E/R164E/K166E (4 K/Rmut) mutations in the linker (Taf14$_{Linker}$) decreased binding of Taf14$_{FL}$ to DNA (Fig. 3c and Supplementary Fig. 4d). We concluded that these two positively charged motifs in Taf14$_{Linker}$ are responsible for DNA binding of Taf14. Interestingly, when either ET domain or the YEATS domain were removed (Taf14$_{ΔET}$ or Taf14$_{ΔYEATS}$) but Taf14$_{Linker}$ remained intact, the binding of these truncated constructs to DNA was over 6-fold tighter than the binding of Taf14$_{FL}$ as seen in EMSA and NMR experiments (Fig. 3d–f). To explore the DNA sequence selectivity of Taf14, we tested binding of Taf14 to 36 bp AT rich dsDNA and 36 bp GC rich dsDNA by EMSA (Supplementary Fig. 5). As shown in Supplementary Figures 5b, c, Taf14 displayed a weak preference for the AT rich DNA sequence. Overall, these data indicate that Taf14$_{Linker}$ binds to DNA via the two positively charged motifs, however the DNA binding activity is reduced when both Taf14$_{YEATS}$ and Taf14$_{ET}$ are present – likely due to the occlusion of the linker region in Taf14$_{FL}$.

**Interaction of Taf14$_{FL}$ with DNA leads to a conformational rearrangement.** How can Taf14$_{Linker}$ be occluded in Taf14$_{FL}$? We investigated the ternary structure of Taf14$_{FL}$ by NMR, mass spectrometry (MS), MST and SAXS. We found that Taf14$_{ET}$ directly binds to Taf14$_{Linker}$ because CSPs were induced in NMR spectra of $^{15}$N-labeled Taf14$_{ET}$ by the peptide corresponding to the linker region (Fig. 3g). The binding affinity of Taf14$_{ET}$ to the Taf14$_{Linker}$ peptide was found to be 640 μM, as measured by MST (Fig. 3h). We note that due to an increase in apparent concentration of the ligand this interaction must be stronger when Taf14$_{Linker}$ and Taf14$_{ET}$ are linked in Taf14$_{FL}$. MS analysis revealed that both Taf14$_{YEATS}$ and Taf14$_{ET}$ are cross-linked to Taf14$_{Linker}$ and therefore are in close proximity (Fig. 3i, Supplementary Fig. 6a and Supplementary Data 1), however the presence of a short 37 bp DNA in the MS sample substantially reduced the number of cross-linked peptides identified between Taf14$_{YEATS}$ and the rest of Taf14 (Fig. 3j and Supplementary Fig. 6b). The SAXS data were in agreement and further suggested that Taf14$_{FL}$ in the apo state is flexible, as indicated by the Kratky plot, and may exist in several conformations in solution, with a 'close' conformation being in dynamic equilibrium with a more 'open' conformation (Fig. 3k and Supplementary Figs. 7 and 8). When a 37 bp DNA was added to the SAXS sample, the complex became rigid (Fig. 3k), and equilibrium shifted toward the predominantly open, DNA-bound conformation (Fig. 3l). Lastly, the incomplete overlap between $^1$H$^{15}$N HSQC spectra of Taf14$_{FL}$ and the isolated Taf14$_{YEATS}$ and Taf14$_{ET}$ suggested

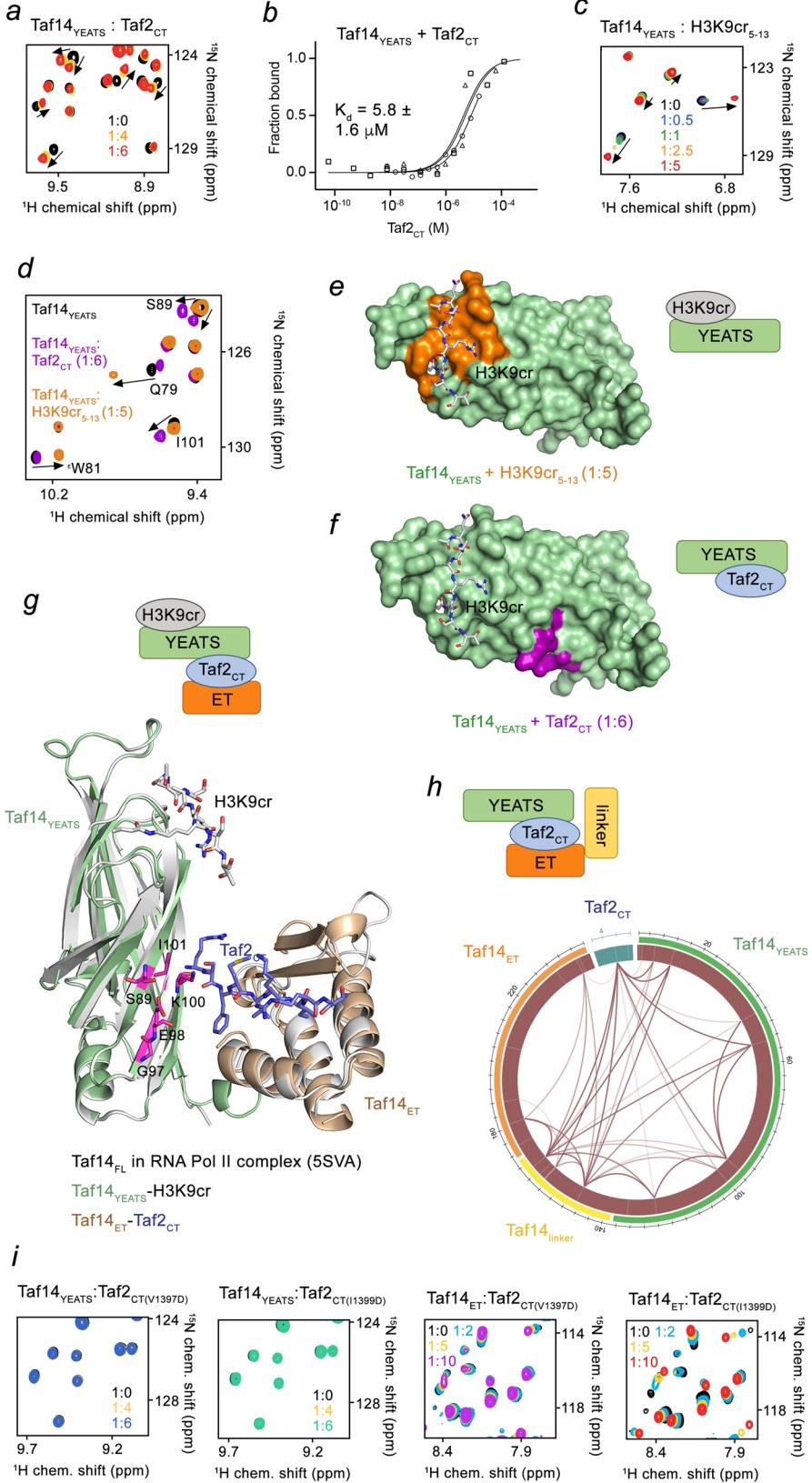

that the domains are not fully independent and interactions or conformational changes occur within Taf14$_{FL}$ (Supplementary Fig. 9). We note that the conformational rearrangement and the ternary structure of Taf14$_{FL}$ did not reduce binding to H3K9cr, indicating that the H3K9cr binding site in Taf14 remains accessible (Fig. 3m).

**Taf2$_{CT}$ promotes binding of Taf14$_{FL}$ to the nucleosome.** Unexpectedly, we found that neither wild type Taf14$_{FL}$ nor the W81A mutant of Taf14$_{FL}$ or Taf14$_{ET}$ appreciably interact with the H3K9cr-containing nucleosome (H3K9cr-NCP) in EMSA experiments (Fig. 4a and Supplementary Fig. 10a–c). In contrast to its binding to the H3K9cr peptide (Fig. 2c)[11], the individual

**Fig. 2 Taf14$_{ET}$ and Taf14$_{YEATS}$ both interact with Taf2$_{CT}$. a** Superimposed $^1$H$^{15}$,N HSQC spectra of Taf14$_{YEATS}$ collected upon titration with Taf2$_{CT}$. The spectra are color coded according to the protein:peptide molar ratio. **b** MST binding curves for the Taf2$_{CT}$ interaction with Taf14$_{YEATS}$. The $K_d$ value represents average of three independent measurements, and error represents the standard deviation. $n = 3$ **c** Superimposed $^1$H$^{15}$,N HSQC spectra of Taf14$_{YEATS}$ collected upon titration with H3K9cr$_{5-13}$ (residues 5–13 of H3) peptide. The spectra are color coded according to the protein:peptide molar ratio. **d** Overlay of $^1$H$^{15}$,N HSQC spectra of Taf14$_{YEATS}$ in the apo state (black) and bound to Taf2$_{CT}$ (purple) or H3K9cr$_{5-13}$ (orange). **e, f** Identification of the H3K9cr- and Taf2$_{CT}$-binding sites of Taf14$_{YEATS}$. Residues that exhibit substantial CSPs in Taf14$_{YEATS}$ upon binding of H3K9cr (e; orange) or Taf2$_{CT}$ (f; magenta) are mapped on the surface of Taf14$_{YEATS}$ (green) with the H3K9cr peptide shown in grey sticks. **g** Overlay of the crystal structures of the Taf14$_{YEATS}$:H3K9cr complex (light green:grey), Taf14$_{ET}$:Taf2$_{CT}$ complex (wheat:purple), and the cryo-EM structure of Taf14$_{FL}$ in the RNA Pol II TFIIF complex (5SVA, grey)[4]. The residues of the Taf2$_{CT}$-binding site of Taf14$_{YEATS}$ are labeled and colored magenta. **h** Mass spectrometry analysis wheel diagram showing positions of DSSO-generated cross-links identified in the Taf14$_{FL}$:Taf2$_{CT}$ complex. Line thickness correlates with the number of cross-link identifications. Taf14 domains and Taf2$_{CT}$ within the circle are colored as in the diagram above and in Fig. 1a. **i** Superimposed $^1$H$^{15}$,N HSQC spectra of Taf14$_{YEATS}$ (two left panels) or Taf14$_{ET}$ (two right panels) collected upon titration with the indicated mutant Taf2$_{CT}$ peptides. Spectra are color coded according to the protein:peptide molar ratio. Schematics of the complexes are shown in (**e, f, g** and **h**).

Taf14$_{YEATS}$ was also incapable of binding to H3K9cr-NCP, reflecting the previous findings that histone tails within the nucleosome are not as freely available for the interactions with readers as peptides (Supplementary Fig. 10b)[16,17]. The association with the nucleosome was detected after the ET domain was removed in Taf14$_{\Delta ET}$ (Fig. 4b) or the YEATS domain was removed in Taf14$_{\Delta YEATS}$ (Supplementary Fig. 10d), indicating that Taf14$_{ET}$ and Taf14$_{YEATS}$ impede the ability of Taf14$_{FL}$ to be engaged with the nucleosome and therefore chromatin. The notable increase in binding to the nucleosome was also observed when Taf2$_{CT}$ was added to the Taf14 samples (Fig. 4c and Supplementary Fig. 10e). The binding of Taf2$_{CT}$ to Taf14$_{FL}$, Taf14$_{\Delta YEATS}$ and Taf14$_{W81A}$, the Taf14 constructs containing the linker region, enhanced the formation of the complexes with the nucleosome. Likewise, binding of Taf14$_{FL}$ to double stranded 37 bp DNA was augmented in the presence of Taf2$_{CT}$ (compare lanes labeled red in Fig. 4d). However, the Taf2-mediated effect in binding to the nucleosome was reduced in the case of Taf14$_{4K/Rmut}$, in which one of the DNA-binding motifs (4K/R) is impaired while another DNA-binding motif (3 K/R) remains (Supplementary Fig. 10f, g). Together, these data demonstrate that the linker region Taf14 is necessary for binding to the nucleosome. The linker region is occluded by Taf14$_{ET}$ and Taf14$_{YEATS}$ in the absence of ligands, whereas binding of the ligand, i.e. Taf2, releases the linker for the binding to DNA and NCP (Fig. 4e). Interestingly, despite the isolated Taf14$_{YEATS}$ did not associate with H3K9cr-NCP (Supplementary Fig. 10b) but bound well to the H3K9cr peptide (Fig. 2c), the interaction of Taf14$_{FL}$ with the unmodified NCP in the presence of Taf2 was enhanced when the crotonylated H3K9 modification was present in the nucleosome (Fig. 4c and Supplementary Fig. 10h, i).

**Association of Taf14 with Taf2 and DNA is essential for transcriptional regulation.** To define the role of the Taf14-Taf2 interaction in Taf14-depended gene regulation, we performed reverse transcriptase (RT) quantitative PCR (RT-qPCR) analysis on a set of genes regulated by Taf14 (*YPR145*, *YER145*, *YKR039W*, and *ACS1*)[18]. *Taf14Δ* cells were transformed with either full-length wild type *TAF14*, a vector only control (to monitor the impact of Taf14 loss), or *taf14* mutants harboring either ET domain truncation (Taf14$_{\Delta ET}$), K149D/R150D/R151D/K161D/K163D/R164D/K166D mutations in the linker region that disrupt binding to DNA (Taf14$_{7K/Rmut}$) or W81A mutation in the YEATS domain that abolishes binding to H3K9acyl (Taf14$_{W81A}$) (Supplementary Fig. 11). As shown in Fig. 4f, deletion of the ET domain resulted in a significant loss of Taf14-mediated suppression of all examined genes. The Taf14$_{7K/Rmut}$ linker mutant also showed a loss of Taf14-mediated suppression similar or equal to the loss of Taf14$_{ET}$ for *YPR145* and *YER145*. The loss of the H3K9acyl binding activity of Taf14 greatly impacted one gene

(*YER145*) but had a minimal or no effect on the others. Taken together, these findings demonstrate a critical role of the Taf2-binding ET domain and the effect of the DNA-binding linker in transcriptional regulation by Taf14.

We next determined the impact of the *taf14* mutations on the recruitment of Taf14 to genes. As shown in Fig. 4g, ChIP-qPCR analysis of the same set of genes examined by RT-qPCR revealed that the loss of the ET domain and the linker region of Taf14 dramatically impaired its recruitment. These results were consistent with the effects imparted by these mutations on gene expression shown in Fig. 4f. In contrast, loss of Taf14 acyl-reading greatly increased the ability of Taf14 to associate with *YER145*, *YKR039W* and *ACS1*. Furthermore, a deletion of *TAF14* (*taf14Δ*) in cellular growth assays resulted in a slow growth phenotype which became more severe at lower temperatures (16 °C and 25 °C) (Fig. 4h). As expected, addition of wild type *TAF14* to *taf14Δ* rescued the observed growth defects, however the rescue effect at lower temperatures was diminished in *taf14Δ* cells expressing Taf14 mutants. Among the mutants, Taf14$_{7K/Rmut}$ and Taf14$_{W81A}$ rescued the growth defects of the *taf14Δ* cells to the highest degree but loss of the ET domain (Taf14$_{\Delta ET}$) led to only a partial rescue of the growth defects, supporting previous findings[19] and pointing to the essential role of this domain in multifaceted functions of Taf14. While the DNA-binding activity of Taf14 was largely dispensable for the yeast growth in the spot assays (Fig. 4h), the Taf14$_{7K/Rmut}$ was deficient in repression of Taf14-mediated suppression of a set of genes (Fig. 4f). The observed differential effects indicate that distinctive functions or a different degree or these functions of Taf14 are necessary for these processes. As Taf14 is a component of multiple complexes, differential effects can also arise depending on which and to what extent the Taf14-containing complexes dominate in these processes.

To assess the importance of Taf2$_{CT}$ for biological activities of Taf2, we examined growth and viability of the Taf2$_{\Delta C}$ mutant in which the C-terminus (residues 1261-1407 of Taf2) was deleted in a plasmid shuffle assay. The Taf2$_{\Delta C}$ variant was shown to disrupt the ability of Taf14 to stably associate with TFIID[14], and it displayed a mild slow growth phenotype at all temperatures tested (Fig. 4i). Notably, additionally mutating residues 775–780 (RRLRKR) of Taf2 to alanine, which were proposed to play a role in DNA binding of Taf2[14], in the Taf2$_{mut\_\Delta C}$ variant led to a strong slow growth phenotype at 25 °C and 30 °C and a temperature sensitive phenotype at 37 °C (Fig. 4i). Furthermore, steady-state mRNA levels of the TFIID-dominated ribosomal protein genes *RPS5*, *RPS3*, *RPS8a* and *RPS9b* in Taf2$_{mut\_\Delta C}$ cells were 6- to 10-fold lower than mRNA levels of these genes in the wild type cells, while steady-state mRNA levels of the glycolytic SAGA dominated genes *PGK1* and *PYK1* were 2-fold lower than mRNA levels of these genes in the wild type cells (Fig. 4j). Finally, Taf2$_{mut\_\Delta C}$ efficiently precipitated TFIID subunits, including

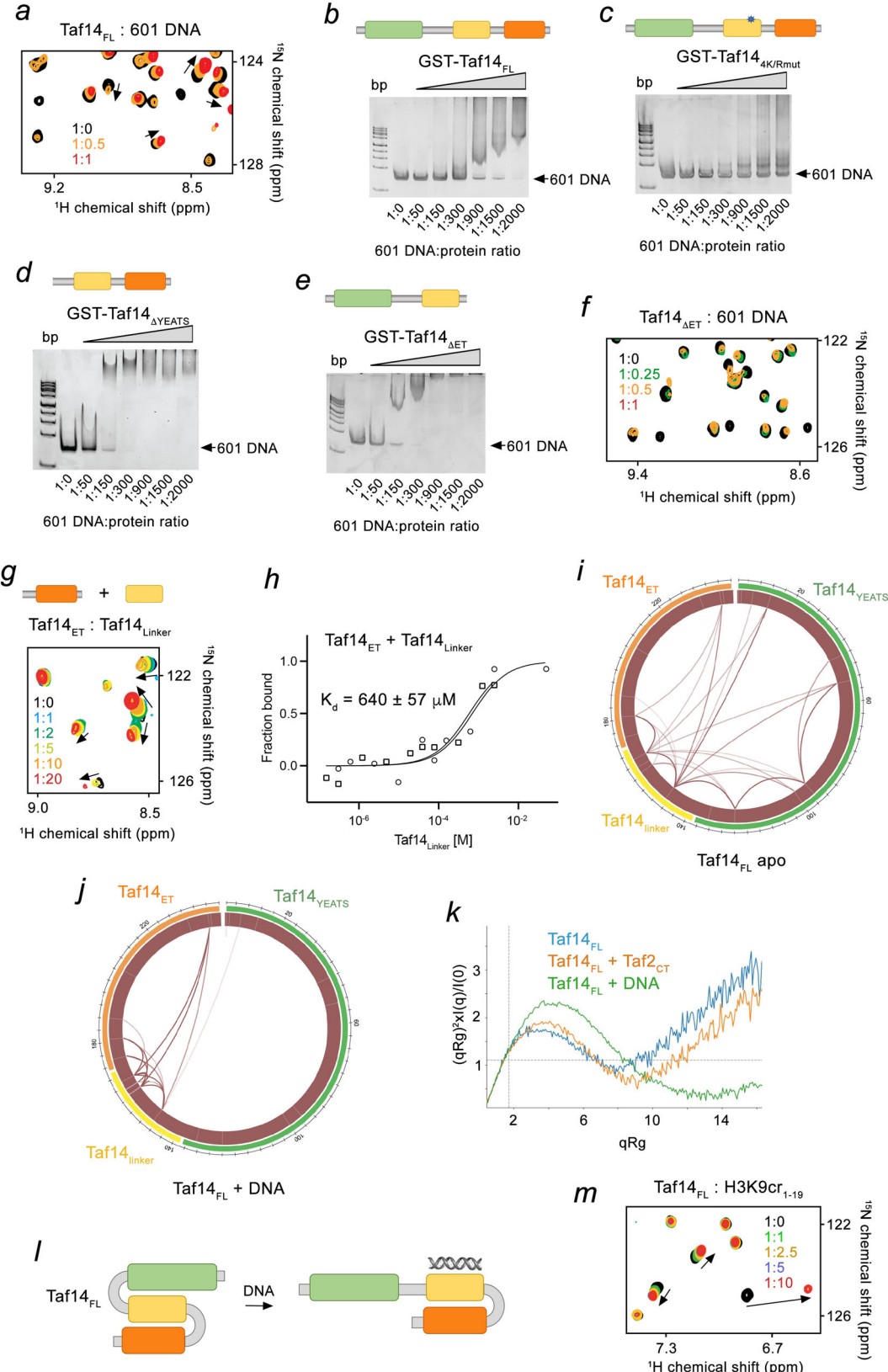

Taf3, Taf7 and Taf8 but not Taf14, in co-IP assays (Fig. 4k). Collectively, these results suggest that the Taf14-dependent and Taf2-dependent interactions with promoter DNA provide a synergistic effect substantially impacting both growth and steady state RNA levels and therefore are important for the regulation of gene transcription in vivo.

In conclusion, our findings demonstrate that Taf14 exists in an autoinhibited state in the absence of binding partners and is unable to bind to the nucleosome, and hence chromatin, even when H3K9acyl is present. Binding of Taf2 to the ET domain and the YEATS domain of Taf14 eliminates the autoinhibition: it causes a substantial rearrangement of Taf14, resulting in

**Fig. 3 DNA binding activity of Taf14$_{Linker}$. a** Overlay of $^1H^{15}N$ HSQC spectra of Taf14$_{FL}$ in the presence of increasing amount of 147 bp 601 DNA. Spectra are color coded according to the protein:DNA molar ratio. **b**–**e** EMSAs of 147 bp 601 DNA in the presence of increasing amounts of the indicated GST-Taf14 proteins. DNA:protein ratio is shown below gel images. Images are from single experiment. Architecture of each Taf14 construct is depicted above the gels, and mutations are indicated with a blue asterisk. **f**, **g** Superimposed $^1H^{15}N$ HSQC spectra of the indicated Taf14 proteins collected upon titration with 147 bp 601 DNA (f) or Taf2$_{Linker}$ **g**. Spectra are colored according to the protein:ligand molar ratio. **h** MST binding curves for the Taf2$_{Linker}$ interaction with Taf14$_{ET}$. The $K_d$ value represents average of two independent measurements, and error represents the standard deviation. $n = 2$ **i**, **j** Mass spectrometry analysis wheel diagram showing positions of DSSO-generated cross-links identified in the Taf14$_{FL}$ (**i**) or Taf14$_{FL}$ + 37 bp dsDNA **j**. Line thickness relates to the number of cross-link identifications. Taf14 domains within the circle are colored as in Fig. 1a. **k** Dimentionless Kratky plots of SAXS analysis for the indicated samples: Taf14$_{FL}$ (blue), Taf14$_{FL}$ + Taf2$_{CT}$ (orange), or Taf14$_{FL}$ + 37 bp dsDNA (green). **l** A model of the conformational rearrangement in Taf14$_{FL}$ upon binding to DNA. **m** Overlay of $^1H^{15}N$ HSQC spectra of Taf14$_{FL}$ in the presence of increasing amounts of H3K9cr$_{1-19}$ peptide. Spectra are colored according to the protein:peptide molar ratio. Source data are provided in a Source Data file.

the release of the linker region which binds to DNA via the positively charged motifs. Our data further suggest that Taf14 may play a role in TFIID promoter recognition in *S. cerevisiae* by concurrently targeting the specific chromatin state (H3K9acyl) and nucleosomal DNA. The two functions of Taf14 may serve to position TFIID for Taf-DNA interactions, and this positioning effect can lead to appropriate deposition of TATA-binding protein (TBP) at the TATA-box or TATA-like element. Indeed, ChIP-exo studies characterizing the positioning of TFIID on promoters genome wide, and independently ChEC-seq studies, have shown that TFIID/TBP positioning in TFIID dominated genes depends on the engagement with the nucleosome[20,21].

This work provides a model for promoter recognition by yeast TFIID at least in part through the histone mark-binding and nucleosomal DNA-binding activities of the Taf14 subunit, which in turn is mediated by another TFIID subunit, Taf2. The identification of the DNA-binding function of Taf14 is important. It has been shown that the Taf1 and Taf2 subunits in metazoan TFIID interact with the initiator (INR) element and the downstream promoter element (DPE)[22], but the DPE and INR have not been identified or tested experimentally in *S. cerevisiae*. Although both metazoan and yeast TFIID contain TBP and can recognize the TATA-box, this sequence is often quite degenerate in promoters and requires additional contacts afforded by Taf subunits. These contacts could involve binding of bromodomains of Taf1 to acetylated histones and/or a PHD finger of Taf3 to H3K4me3 in metazoan TFIID and binding of the YEATS domain of Taf14 to H3K9acyl in yeast TFIID[9,11,23–26]. Lastly, because the Taf2$_{mut\_\Delta C}$ variant led to a strong slow growth phenotype, the functional importance of the region encompassing residues 775–780 of Taf2[14] needs to be examined in future studies. Our study serves as benchmark for further analysis of the yeast TFIID engagement with chromatin.

## Methods

**Protein purification**. The *S. cerevisiae* Taf14$_{FL}$ (aa 1–244), Taf14$_{YEATS}$ (aa 1–137), Taf14$_{\Delta ET}$ (aa 1–174), Taf14$_{\Delta YEATS}$ (aa 148–243) and Taf14$_{ET}$ (aa 162–243, aa 168–243) constructs were cloned into pGEX-6P1, pET22b, pDEST15 or pDEST17 vectors. The Taf14$_{3K/R}$ (K149E/R150E/R151E), Taf14$_{4K/R}$ (K161E/K163E/R164E/K166E), and Taf14$_{ET}$ construct for NMR (K163N/R164Q/K166N) were generated using the Stratagene QuikChange Lightning Site-Directed Mutagenesis Kit. Sequences were confirmed by DNA sequencing. Taf14$_{ET}$ (aa 168–242) construct was used for all experiments except NMR, where the Taf14$_{ET}$ (aa 162–242 K163N/R164Q/K166N) construct was used. All proteins were expressed in *Escherichia coli* Rosetta-2 (DE3) pLysS cells grown in either Terrific Broth, Luria Broth or in minimal media supplemented with $^{15}NH_4Cl$ (Sigma). Protein production was induced with 0.3 mM IPTG for 18 h at 16 °C. Bacteria were harvested by centrifugation and lysed by sonication. GST-fusion proteins were purified on glutathione agarose 4B beads (Thermo Fisher Sci). The GST-tag was cleaved with either PreScission or tobacco etch virus (TEV) protease, or the fusion proteins were eluted from the resin with reduced L-glutathione (Fisher). His-fusion proteins were incubated with Ni-NTA resin (ThermoFisher), washed and eluted with a gradient of imidazole. When necessary, removal of the His-tag was completed during an overnight dialysis, while incubating the protein with TEV protease. Proteins were further purified by column chromatography and concentrated in

Millipore concentrators (Millipore). Proteins used in this study are summarized in the source data file.

**DNA purification and preparation**. Double stranded DNA containing the 147 bp 601 Widom DNA sequence in the pJ201 plasmid (32x) was transformed into DH5α cells. The plasmids containing the 601 sequence were amplified in TB media and purified using the PureLink HiPure Expi Plasmid Gigaprep Kit (Invitrogen K210009XP). Separation of the individual sequences was completed by digestion of the plasmid with EcoRV (NEB) followed by PEG and ethanol precipitation. After the final purification step, the DNA was suspended in water and the 601 DNA sequences were evaluated for purity by native polyacrylamide gel electrophoresis.

The double stranded DNA for EMSA and mass spectrometry were purchased as single stranded DNA (IDT) and annealed. DNA sequences used in this study are listed in the source data file. Complimentary DNA strands were combined in a 1:1 molar ratio in water in PCR tubes. Using a thermocycler to regulate temperature, the samples were brought to 95 °C for 20 min and then to 16 °C at a rate of 0.1 °C/second. All samples of annealed DNA were evaluated for purity by native polyacrylamide gel electrophoresis.

**H3K9cr-nucleosome preparation and assembly**. Recombinant Histone His-TEV-H2A, His-TEV-H2B, His-SUMO-TEV-H4 and H3K9cr or H3 were purified as described previously[15]. All histone pellets were dissolved in 6 M GuHCl buffer (6 M guanidinium hydrochloride, 20 mM Tris, 500 mM NaCl, pH 7.5), and concentration was measured by UV absorption at 280 nm (Biotek synergy H1 plate reader). To prepare H2A/H2B dimer, His-TEV-H2A and His-TEV-H2B were mixed in the molar ratio of 1:1, and 6 M GuHCl buffer was added to adjust total protein concentration to 4 µg/µl. Denatured His-TEV-H2A/His-TEV-H2B solution was dialyzed sequentially with stirring at 4 °C against 2 M TE buffer (2 M NaCl, 20 mM Tris, 1 mM EDTA, pH 7.8), 1 M TE buffer (1 M NaCl, 20 mM Tris, 1 mM EDTA), 0.5 M TE buffer (0.5 M NaCl, 20 mM Tris, 1 mM EDTA). The resulting dimer solution was centrifuged for 5 min at 4 °C to remove precipitates, and the concentration of dimer was determined by UV absorption at 280 nm. His-TEV-H3/His-SUMO-TEV-H4 tetramer was refolded as His-TEV-H2A/His-TEV-H2B above except without stirring and the total protein concentration was adjusted to 2 µg/µl prior to dialysis. Then His-TEV-H2A/Hi-TEV-H2B dimers were mixed with His-TEV-H3/His-SUMO-TEV-H4 tetramers in a molar ratio of 1:1 to generate histone octamers, and NaCl solid was added to adjust NaCl concentration to 2 M. The 147 bp biotinylated 601 DNA was prepared by Polymerase chain reaction with biotinylated primers and purified with PCR cleanup kit (#2360250 Epoch Life Science). Purified 147 bp DNA was re-dissolved in 2 M TE buffer and added to histone octamer solution in the molar ratio of 0.85:1. 2 M TE buffer was added to adjust final 147 bp DNA concentration to 2–3 µM. The DNA histone mixture solution was then transferred to a dialysis bag and placed inside about 200 ml 2 M TE buffer, while stirring at room temperature. Tris buffer with no salt (20 mM Tris) was slowly added into the 2 M salt buffer through a liquid transfer pump (#23609-170 VWR®). Nucleosomes formed when salt concentration was reduced to around 150 mM (measured by EX170 salinity meter), and the DNA histone mixture solution was further dialyzed into low salt Tris buffer (20 mM Tris, 20 mM NaCl, 0.5 mM EDTA, pH 7.8). Precipitates were removed by centrifuge, and the nucleosome concentration was measured by the absorbance at 260 nm (A260) using the Biotek synergy H1 plate reader. His-TEV protease was added to nucleosome solutions (TEV:nucleosome 1:30, w-w) to remove all the histone tags after incubation for 1 h at 37 °C, and finally all the His tagged impurities were removed from the nucleosome solution by Ni$^{2+}$-NTA resin (Thermo Fisher #88221).

**EMSA**. EMSAs were performed by mixing increasing amounts of Taf14 proteins with 0.25 pmol/lane of 601 DNA, dsDNA (36 bp AT rich, 36 bp GC rich, 37 bp non-specific), NCP, or H3K9cr-NCP in 20–25 mM Tris-HCl pH 7.5, 100–150 mM NaCl and 0.2 mM ethylenediaminetetraacetic acid (EDTA) (reactions with NCP or H3K9cr-NCP were supplemented with 20% glycerol) in a 10 µL reaction volume. For the samples containing the DNA ladder (O'RangeRuler 5 bp DNA Ladder,

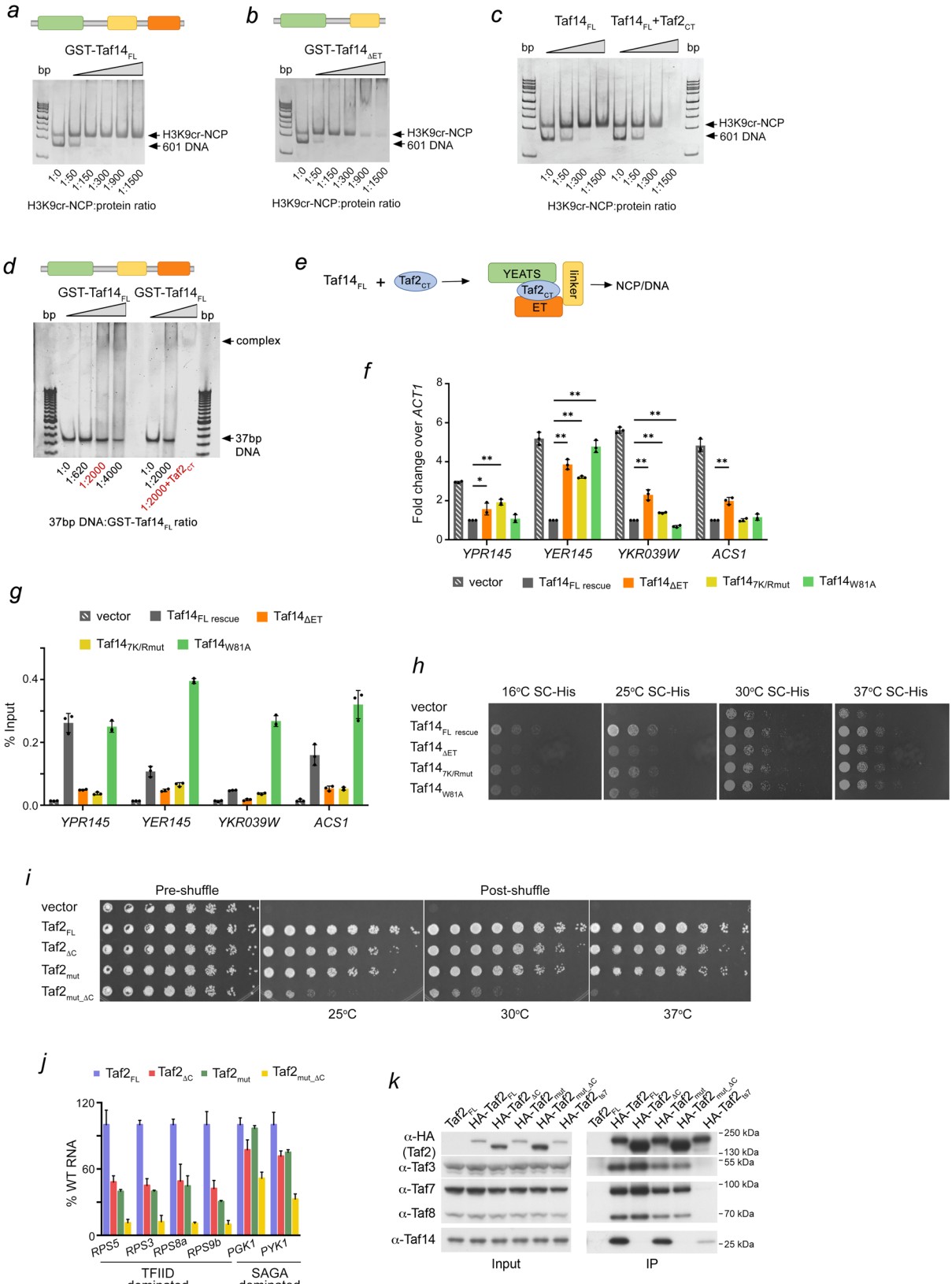

ThermoSci) the buffer was 20 mM Tris-HCl pH 7.5, 150 mM NaCl and 5 mM DTT. The indicated Taf14 constructs were preincubated with Taf2$_{CT}$ at a 1:2 (Fig. 4c, Supplementary Figs. 5d, e, 10e, g, h) or 1:10 (Fig. 4d) molar ratio for a minimum of 1 h on ice prior to addition of DNA or NCP. Reaction mixtures were incubated at 4 °C for 10 min (2–2.5 µl of loading dye was added to each sample) and loaded onto a 5 or 10% native polyacrylamide gel. Electrophoresis was performed in 0.2x Tris-borate-EDTA (TBE) at 80–120 V on ice. The gels were stained with SYBR Gold (Thermo Fisher Sci) and visualized by Blue LED (UltraThin LED Illuminator-GelCompany).

**NMR experiments.** Nuclear magnetic resonance (NMR) experiments were performed at 298 K on Varian INOVA 900 MHz and 600 MHz spectrometers equipped with cryogenic probes. The NMR samples contained 0.1–0.3 mM

**Fig. 4 Binding of Taf14$_{FL}$ to the nucleosome requires binding to Taf2$_{CT}$. a–c** EMSAs of H3K9cr-NCP$_{147}$ in the presence of increasing amounts of the indicated GST-Taf14 or Taf14 proteins (and Taf2$_{CT}$ in c). Architecture of each Taf14 construct is depicted above gels. Images are from single experiment. **d** EMSA of 37 bp dsDNA in the presence of increasing amounts of the indicated GST-Taf14 (and Taf2$_{CT}$ in lane 9). Architecture of the Taf14 construct is depicted above the gel. GST-Taf14$_{FL}$ and Taf2$_{CT}$ were premixed at a 1:10 molar ratio prior to incubation with DNA. **e** A schematic of the mechanism for the engagement of Taf14$_{FL}$ with NCP: following the interaction with Taf2$_{CT}$, the released Taf14$_{Linker}$ binds DNA/NCP. **f** Reverse transcriptase qPCR analysis of various transcripts in the *TAF14* full-length rescued strain, vector only strain, *taf14* ET delete strain (Taf14$_{\Delta ET}$), and *taf14* mutant strains (Taf14$_{7k/Rmut}$ and Taf14$_{W81A}$). The mean ± SD are calculated from three biological replicates. *ACT1* was used as a normalization control. Two-tailed multiple T tests were performed, *$P < 0.05$; ** $P < 0.005$. **g** Chromatin immunoprecipitation assays for promoter-bound Taf14 in the same strains and at the same set of genes examined in (f). The mean ± SD are calculated from three biological replicates. **h** Yeast spotting assays with wild type (BY4741) or *taf14Δ* strains transformed with the indicated plasmids and grown on SC-His medium at the indicated temperatures. **i** Plasmid shuffle assay. *Taf2* null cells transformed with wild type *TAF2* or LEU2-marked mutant *taf2*-containing variants plasmids were plated onto SC-Leu (pre-shuffle) or SC-Leu + 5-FOA (post-shuffle). Plate images are representative of at least two biological replicates. **j** Steady-state RNA analysis. qRT-PCR was performed on total yeast RNA. Data is presented as the ratio of indicated target gene/U3 with wild type *TAF2* set as 100%. Each pRT-PCR reaction was performed in triplicate using three biological experiments, with bars representing mean and errors representing standard deviation. **k** Co-IP of the HA-Taf2 proteins with the indicated TFIID proteins was analyzed by immunoblotting using anti-HA and anti-Taf polyclonal antibodies. Representative immunoblots from two biological replicates are shown. Source data are provided in a Source Data file.

---

uniformly $^{15}$N-labeled Taf14 (Taf14$_{FL}$, Taf14$_{YEATS}$, Taf14$_{\Delta ET}$, or Taf14$_{ET}$ (aa 162–243 K163N/R164Q/K166N)) in 50 mM Tris-HCl pH 6.6–7.5, 150 mM NaCl, 0–5 mM DTT or 0.2 mM TCEP or 1x PBS pH 6.6–7.5 (supplemented with up to an additional 100 mM NaCl), 0–5 mM DTT and 8–10% D$_2$O. Binding was characterized by monitoring chemical shift changes in $^1$H$^{15}$,N HSQC spectra of the proteins induced by the addition of Taf2$_{CT}$ (aa 1392–1402), Taf2$_{CT(V1397D)}$ (aa 1392–1402), Taf2$_{CT(I1399D)}$ (aa 1392–1402), Taf14$_{Linker}$ (aa 145–169), or H3K9cr$_{5-13}$ peptides (all peptides synthesized by Synpeptide) or 601 DNA. NMR data were processed and analyzed using NMRPipe as previously described[27]. NMR assignments of Taf14$_{YEATS}$ were taken from[28].

**X-ray crystallography.** Taf14$_{ET}$ was concentrated to ~3.5 mM (in 50 mM Tris-HCl pH 7.5, 150 mM NaCl, 0.2 mM TCEP). The protein was incubated with the Taf2 peptide (Synpeptide, aa 1392–1402) at a molar ratio of 1:2.5 for 1 h. The protein:peptide solution was concentrated to ~ 6.6 mg/mL in buffer (20 mM Tris-HCl pH 7.5, 100 mM NaCl) for crystallization. Crystals were grown using sitting-drop vapor diffusion method at 23 °C by mixing 0.6 μL of protein with 0.6 μL of well solution composed of 2.4 M sodium malonate pH 7.0. Crystals were cryo-protected with 30% (v/v) glycerol. X-ray diffraction data were collected on a beamline 4.2.2 at Advance Light Source. Indexing and scaling were completed using HKL2000[29]. The structure was determined using Phenix phaser with the Taf14$_{ET}$ (PDH ID: 6LQZ) structure as a search model. Model building was carried out with Coot[30], and refinement was performed with Phenix refine[31]. Crystallographic statistics for the Taf14$_{ET}$:Taf2 structure are shown in Supplementary Table 1.

**MST.** Microscale thermophoresis (MST) experiments were performed using a Monolith NT.115 instrument (NanoTemper). All experiments were performed using SEC purified Taf14 proteins (His-Taf14$_{FL}$, His-Taf14$_{ET}$, His-Taf14$_{YEATS}$, or Taf14$_{ET}$) in a buffer containing 20 mM Tris-HCl (pH 7.5), 150 mM NaCl and 0–0.2 mM TCEP. For the Taf14$_{ET}$:Taf2$_{CT}$ interaction, the concentration of fluorophore, FAM-labeled Taf2$_{CT}$ peptide (Synpeptide, 1392–1703 aa), was 20 nM. To monitor binding between Taf2$_{CT}$ and Taf14$_{FL}$ or Taf14$_{YEATS}$, and Taf14$_{Linker}$ and Taf14$_{ET}$, His-tagged proteins were labeled using a His-Tag Labeling Kit RED-tris-NTA (2$^{nd}$ Generation, Nanotemper), and the concentration of each labeled protein was kept at 20 nM. Dissociation constants were determined using a direct binding assay in which increasing amounts of unlabeled binding partner (Taf2$_{CT}$ peptide or Taf14$_{Linker}$ peptide) were added stepwise. The measurements were performed at 80% LED and medium MST power with 3 s steady state, up to 20 s laser on time and 1 s off time. The K$_d$ values were calculated using MO Affinity Analysis software (NanoTemper) ($n = 2$ or 3). Figures were generated in GraphPad PRISM.

**Sample preparation for cross-linking mass spectrometry.** Each sample contained 1 mM SEC purified Taf14$_{FL}$, without ligands or with 37 bp double stranded DNA or Taf2$_{CT}$ peptide (aa 1392–1403) (9.6 and 10.8 mM respectively) in 25 mM Hepes pH 7.5, 150 mM NaCl for 10 min prior to the addition of 2.5 mM DSSO (or DMSO for control reaction). The samples were incubated for 1 h at 23 °C, followed by the addition of 20 mM Tris-HCl pH 7.5 for 10 min to stop the reaction. DSSO cross-linked Taf14$_{FL}$ samples were denatured, reduced and alkylated with 5% (w/v) SDS, 10 mM tris(2-carboxyethylphosphine) (TCEP), 40 mM chloroacetamide, 50 mM Tris-HCl, pH 8.5 boiling 10 min. The lysates were cleared via centrifugation at 13,000 x g for 10 min at ambient temperature. Cross-linked Taf14$_{FL}$ samples were digested using the SP3 method[32]. Carboxylate-functionalized speedbeads (Cytiva Life Sciences) were added. Addition of acetonitrile to 80% (v/v) caused the proteins to bind to the beads. The beads were washed twice with 80% (v/v) ethanol

and twice with 100% acetonitrile. Samples were digested in 50 mM Tris buffer, pH 8.5, with 1 μg Lys-C/Trypsin (Promega) incubated at 37 °C overnight rotating to mix. Tryptic peptides were desalted using the carboxylate-functionalized speed-beads with the addition of 95% (v/v) acetonitrile twice and then eluted with 3% (v/v) acetonitrile, 1% (v/v) trifluoroacetic acid and dried in a speedvac vacuum centrifuge.

**MS analysis of DSSO cross-linked proteins.** Samples were suspended in 3% (v/v) acetonitrile/0.1% (v/v) trifluoroacetic acid and approximately 2 picomoles of tryptic peptides were directly injected onto a reversed-phase C18 1.7 μm, 130 A˚, 75 mm × 250 mm M-class column (Waters), using an Ultimate 3000 UPLC (Thermo Scientific). Peptides were eluted at 300 nL/minute using a gradient from 2% to 20% acetonitrile over 40 min and detected using a Q-Exactive HF-X mass spectrometer (Thermo Scientific). Precursor mass spectra (MS1) were acquired at a resolution of 120,000 from 350 to 1600 m/z with 25 eV in-source collision induced dissociation, an AGC target of 3E6 and a maximum injection time of 50 ms. Dynamic exclusion was set for 5 seconds with a mass tolerance of ±10 ppm. Precursor peptide ion isolation width for MS2 fragment scans was 1.4 Da, and the top 15 most intense ions were sequenced. All MS2 sequencing was performed using higher energy collision dissociation (HCD) at 30% normalized collision energy. An AGC target of 1E5 and 100 ms maximum injection time was used. Raw files were searched against the Uniprot yeast sequences for Taf14 (accession P35189) using pLink2 with the enzyme specificity set to trypsin with two missed cleavages. The peptide mass range and length were 500–6000 and 5–60, respectively. Precursor and fragment tolerance were both set to ±20 ppm. Final results were filtered with the MS1 tolerance set to ±8 ppm. FDR was set to ≤1% at spectral level and E-values were calculated. Using a proXL conversion script (https://github.com/yeastrc/proxl-import-plink2), xml files were generated and submitted to the proXL website. The crosslinking MS circle diagrams were generated using proXL and the data has been deposited and is available through the proXL web interface at https://proxl-ms.org/.

**SAXS.** Small angle X-ray scattering (SAXS) experiments were performed at beamline 12-ID-B, the Advanced Photon Sources facility (Argonne National Laboratory, Lemont, IL). Photon energy was 13.3-keV (wavelength λ was 0.932 Å). Scattered x-ray intensities were measured using a Pilatus 2 M detector. A sample-to-detector distance of 1.9 m was used to obtain $q$ range of $0.005 < q < 0.88$ Å$^{-1}$. Here $q$ is magnitude of the scattering vector, $q = (4\pi/\lambda)\sin\theta$, where $2\theta$ is the scattering angle and $\lambda$ is the wavelength of the radiation. The concentration series measurements were carried out to remove the scattering contribution due to interparticle interactions and to extrapolate the data to infinite dilution. The concentrations of Taf14$_{FL}$ were 2.7, 3.6, 7.2 μM in a buffer consisting of 50 mM HEPES, 150 mM NaCl, 0.2 mM TCEP pH 7.5. For the solution of the complex, the concentration of Taf14$_{FL}$ was the same, with the molar ratio of Taf14$_{FL}$ to ligand (Taf2$_{CT}$ peptide or DNA) of 1:4. The ligand (concentrations of 8.1, 10.8, 21.6 μM) in buffer were used as background for the complex. The solutions were loaded in a flow cell made of a cylindrical quartz capillary (1.5-mm diameter and 10-μm wall-thickness) for the measurements. To minimize radiation damage and obtain a good signal-to-noise ratio, 45 images were taken for each sample and buffer with an exposure time of 0.5 s. The two-dimensional scattering images were converted to one-dimensional SAXS curves through radial averaging after solid angle correction and then normalizing with the intensity of the transmitted x-ray beam, then were averaged after elimination of outliers using the software package developed at beamline 12-ID-B.

The buffer background subtraction and intensity extrapolation to infinite dilution were carried out using MatLab script. The radius of gyration Rg and

intensity at zero angle I(0) were generated from Guinier plot in the range of $qR_g < 1.3$. For comparison, Rg and I(0) were also calculated in real and reciprocal spaces using program GNOM in q range up to 0.30 Å$^{-1}$. The pair-distance distribution function P(r) and maximum dimension ($D_{max}$) were also calculated using GNOM. In the process of producing $P(r)$, the $D_{max}$ was chosen in the way that the $P(r)$ curves approach zero at $D_{max}$ in a smooth concave manner. The molecular weights were estimated based on Bayesian statistics method. The obtained structural parameters and molecular weights are shown in Supplementary Fig. 4a, and Guinier plots are shown in Supplementary Figs. 4c and 5c. The Rg based dimensionless Kratky plots presented as $(qRg)2I(q)/I(0)$ vs. $qRg$ was used for analysis of flexibility and/or degree of unfolding in samples[33].

**Spotting assays**. Spotting assays for wild type and mutated Taf14 were performed with serial dilutions of saturated overnight cultures of the indicated yeast strains, and growth was assessed after 2–3 d (dilution ratio of 1:5). All yeast strains used in this study are described in Supplementary Table 2, and plasmids used in this study are presented in Supplementary Table 3.

**qPCR**. For Taf14: reverse transcriptase qPCR was performed as described previously[15]. *taf14Δ* cells transformed with either vector only, *TAF14* full-length or the indicated *taf14* mutants were cultured in rich media YPD overnight at 30 °C. Ten OD$_{600}$ of cell pellets were collected from each strain and phenol/chloroform extraction was used to extract RNA. Residual DNA was digested by DNase I. Complementary DNA (cDNA) was produced from 2 mg of extracted RNA using IScript kit (BioRad). cDNA was then diluted to 0.1 mg/ml for reverse transcriptase qPCR using SYBR green. *ACT1* was used as the normalization control.

For Taf2: For RNA analysis, total RNA was subjected to reverse transcription with oligo(dT) and a gene specific U3 primer. qPCR was performed with SYBR green supermix (Bio-Rad). Samples were quantified using the relative standard curve method, normalized to U3, and expressed as a percentage of the average of the *HA-TAF2* strain. All qPCR reactions were performed in triplicate. Two biological replicates were used for each sample.

**Chromatin immunoprecipitation (ChIP)**. ChIP assays were performed as previously described[34]. *taf14Δ* cells transformed with either vector only, *TAF14* full-length or the indicated *taf14* mutants were cultured in rich media YPD overnight at 30 °C. Fifty OD$_{600}$ of cell pellets were collected from each strain and fixed in 1% formaldehyde for 15 min prior to processing for chromatin immunoprecipitation using a rabbit antibody directed against Taf14 (gift from Joseph Reese), 5 µl per 500 µl reaction. Primers directed against the gene regions are shown in the source data file.

**Plasmid shuffle assay**. The assay was performed essentially as described[14]. Briefly, a taf2 null strain (JFTAF2del) was used for all experiments. For plasmid shuffle analyses, JFTAF2del was transformed an empty p415ADH plasmid, a p415ADH-*TAF2* plasmid, a p415ADH-*HA$_{x3}$NLS-TAF2* plasmid, or a p415ADH-*HA$_{x3}$NLS-TAF2* mutant plasmid, and transformants were grown on SC − Leu plates. Colonies were grown to saturation in SC − Leu at 30 °C prior to spotting and grown at various temperatures between 20 °C and 37 °C (cell dilution ratio of 1:4).

**Co-IP**. Co-immunoprecipitations were performed using pseudodiploids containing both a wild type *URA3*-marked *TAF2* gene and a WT or mutant *LEU2*-marked *TAF2* gene. All variants indicated with HA include a 3xHA-NLS N-terminal tag. Two mg of soluble protein extract were incubated with 2.5 µg of anti-HA 12CA5 mAb and 50 ng/µl ethidium bromide in a total volume of 412 µl at 4 °C for 2 h. Immune complexes were captured with 10 µl of protein A-Sepharose beads (Life Technologies), separated via SDS-PAGE and electrotransferred. Proteins were detected with anti-HA and anti-Taf polyclonal antibodies, followed by anti-rabbit Fc HRP-conjugated secondary antibodies (Abcam, 1:10,000 dilution) for chemiluminescent detection. Antibodies: anti-HA blot 3F10 (Sigma-Millipore, cat # 12013819001, 1:2000 dilution), anti-HA IP 12CA5 (made in house, 2.5 µg per reaction), anti-Taf3 (made in house, 1:1000 dilution), anti-Taf7 (made in house, 1:2000 dilution), anti-Taf8 (made in house, 1:1000 dilution), anti-Taf14 (made in house, 1:2000 dilution).

**Reporting summary**. Further information on research design is available in the Nature Research Reporting Summary linked to this article.

## Data availability

The data that support this study are available from the corresponding author upon reasonable request. Coordinates and structure factors of the Taf14 ET-Taf2 CT complex have been deposited in the Protein Data Bank under the accession code 7UHE. The mass spec data have been deposited to the PRIDE database under accession number PXD033397. Source data are provided with this paper.

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

## Acknowledgements

We thank Xiaobing Zuo for help with SAXS experiments. This work was supported in part by grants from the NIH: HL151334, GM125195, GM135671, CA252707 and AG067664 to T.G.K., GM49985 and DK121366 to J.T.F., GM126900 to B.D.S., GM127575 to W.R.L. and from Welch Foundation: A-1715 to W.R.L. We acknowledge the use of the SAXS Core facility of Center for Cancer Research (CCR), National Cancer Institute (NCI). This project has been funded in part with federal funds from the National Cancer Institute, NIH, under contract HHSN26120080001E. The content of this publication does not necessarily reflect the views or policies of the Department of Health and Human Services, nor does mention of trade names, commercial products, or organizations imply endorsement by the U.S. Government. This research was supported in part by the Intramural Research Program of the NIH, National Cancer Institute, Center for Cancer Research. This research used 12-ID-B beamline of the Advanced Photon Source, a U.S. Department of Energy (DOE) Office of Science User Facility operated for the DOE Office of Science by Argonne National Laboratory under Contract No. DE-AC02-06CH11357.

## Author contributions

B.J.K., J.T.F., J.Z., C.C.E., L.F., R.K.S. and W.W.W. performed experiments and together with L.R.S., T.L., K.C.H., W.R.L., Y.-X.W., B.D.S., P.A.W. and T.G.K. analyzed the data. B.J.K., J.T.F. and T.G.K. wrote the manuscript with input from all authors.

## Competing interests

The authors declare no competing interests.
