## [Peer Review File · Nature Communications]

REVIEWER COMMENTS

Reviewer #1 (Remarks to the Author):

This manuscript demonstrates the structural basis of the Taf14-Taf2 interaction and proposes a plausible model that Taf2 modulates Taf14 conformation to enhance Taf14's interactions with nucleosomes. In this study, the authors first determined the crystal structure of the Taf14ET-Taf2CT structure. Surprisingly, they found that besides Taf14ET, Taf14YEATS can also bind Taf2CT. Cross-link mass spectrometry and NMR analyses identified a potential binding interface between Taf14YEATS and Taf2CT. More importantly, they discover that Taf2CT binding induces the structural arrangement of Taf14FL, which exposes the linker region to bind DNA and nucleosome. Some in vivo studies confirm the importance of Taf14-Taf2 and Taf14-DNA interactions in Taf14-mediated transcription regulation. Together, this study presents an impressive amount of work identifying the molecular mechanism of the Taf14-Taf2 interaction and its possible role in modulating the Taf14-NCP interaction. However, this manuscript has several significant weaknesses, and the present data doesn't fully support the authors' conclusions. Here I list some major concerns in each Result section.

1. Structural insight into the Taf14ET-Taf2CT complex

The identity of Taf2CT is never defined in the main text. In the Methods section, Taf2CT has been referred to as 1392-1402 or 1392-1403. How do authors determine the minimum fragment of Taf2CT required for Taf14ET interaction? If cited from a previous publication, the appropriate reference is needed.

2. Both Taf14ET and Taf14YEATS bind to Taf2CT

This result is quite unexpected and contradicts the coauthors' previous publication (J Biol Chem. 2016, 291:22721-22740). In the 2016 JBC paper, GST pull-down assay showed Taf14YEATS did not bind Taf2. Thus, this discrepancy needs to be explained. How Taf14ET binds Taf2CT remains unsolved. The structural model (Fig. 2g) and related biochemical data are not self-consistent. First, according to the structural model (Fig. 2g), Taf2CT must anchor to Taf14ET, and then the N-terminal 2-3 residues of Taf14CT make contacts with Taf14YEATS. It is very unlikely that Taf14YEATS (in the absence of Taf14ET) binds Taf2CT using the same contacting interface. Such a small contacting Taf14YEATS-Taf2CT interface, as seen in Fig. 2g, cannot support the strong binding between Taf14YEATS and Taf2CT (Kd: 6 μ M). Second, Taf2 V1397D and I1399D completely disrupt the interaction between Taf14YEATS and Taf2CT, but this is not supported by the proposed structural model in which Taf2 V1397 and I1399 make no contact with Taf14YEATS. Third, Taf2 bridges two Taf14 domains together to favor a closed conformation but not an "open" conformation in this structural model. It contradicts the authors' discussion (line 213-215) "Binding of Taf2 to the ET domain and the YEATS domain of Taf14 eliminates the autoinhibition: it causes a substantial rearrangement of Taf14, resulting in the release of the linker region which binds to DNA via the positively charged motifs".

3. Taf14ET and Taf14YEATS regulate DNA binding activity of the linker (Taf14Linker)

This finding is novel but needs more controls. Can Taf14ET or Taf14YEATS alone bind DNA? Can Taf14Linker bind DNA without both Taf14ET and Taf14YEATS? Is there any DNA sequence preference or specificity for Taf14? I think these are critical controls. It is also better to quantify the binding affinity (or Kd) of the Taf14-DNA interaction to show how significant this interaction is.

4. Interaction of Taf14FL with DNA leads to a conformational rearrangement

The evidence for conformational changes comes from NMR, MS, and SAXS. Although I am not an expert on SAXS, I still have several concerns about SAXS data. First, the data quality shown by the Guinier plot (Supplementary Fig. 4c) is low, indicating the conclusion from the apo Taf14FL is not reliable. Second, the Kratky plots of apo Taf14FL and Taf14FL+Taf2CT indicate a partially unfolded structure, not just "flexible" or "several conformations in equilibrium." Third, the SAXS data is not fully consistent with the role of

Taf2CT in inducing a conformational change of Taf14. Authors proposed that Taf2CT binding Taf14 could induce Taf14 rearrangement to form the "open" conformation to enable its interaction with DNA, but the SAXS data showed that Taf14FL and Taf14FL+Taf2CT have similar SAXS curves.

5. Taf2CT promotes binding of Taf14FL to the nucleosome

This is an important finding. Again, more critical controls should be included to reveal how Taf2CT promotes the binding of Taf14FL to the nucleosome. First, Taf14 Δ YEATS, as well as Taf14YEATS and Taf14ET alone, should be tested by the similar EMSA as Taf14 Δ ET (Fig.4b) to see which Taf14 domain is important for NCP binding. Second, WT NCP should be tested to check whether the Cr modification plays a dominant role in the Taf14-NCP interaction or the binding is dominantly determined by the Taf14-DNA interaction. Third, considering Taf14ET could bind a lot of short motifs conserved as Taf2CT, will other Taf14-binding peptides, e.g., Sth1 peptide, have the similar effect of promoting the Taf14-H3K9cr NCP binding?

6. Association of Taf14 with Taf2 and DNA is essential for transcriptional regulation

Fig. 4f: First, the control genes YLR290 and YAL017 should be included in the figure. Second, the significance of the differences between bars should be shown. Third, I don't see much difference between Taf14 W81A and 7K/R mutant (only show the difference on two genes out of 12 tested genes). It is against the authors' description in lines 183-187. More importantly, to ascertain that association of Taf14 with Taf2 is essential for Taf14-mediated transcriptional regulation, the Taf14 or Taf2 mutants (point mutations instead of truncations) disrupting the Taf14-Taf2 interaction should be examined to show the biological importance of the Taf14-Taf2 interaction directly. The current used Taf14ET is not an appropriate mutant to confirm the role of Taf14-Taf2 interaction in Taf14-mediated transcriptional regulation since Taf14ET is involved in binding with more than ten other proteins. A Taf2 point mutation that disrupts Taf14-Taf2 interaction while not affecting other Taf14-mediated interactions should be tested to see if it affects Taf14-mediated gene expression.

Some typos:

Line 170: should be Suppl. Figs. 2d and 2e

Line 202: should be Fig.4h

Reviewer #2 (Remarks to the Author):

The manuscript entitled "Taf2 mediates DNA binding of Taf14" by Klein et al. proposes the role of Taf2 C-terminal domain in activation of intrinsic DNA binding activity of Taf14 protein. Although the manuscript is well written, its main conclusions are not well supported by the experimental data. Authors systematically mis- and over-interpret the data and also the previously published results. Although the presented story is fascinating, the manuscript needs major revision to provide the data that do support the conclusions and it also needs major revision of the text. My concerns are:

Major issues:

1. In Figure 1C, the authors determine Kd of Taf14-ET binding to Taf2CT. The value is 1.6 and when comparing it to the full-length Taf14-Taf2CT binding (Fig. 1H; Kd=0.7 μ M) the authors state that "... binding of full-length protein is two-fold tighter..." (line 93) emphasizing the importance of this difference. In figure 2B the authors measure the Kd of Taf14-YEATS interaction with Taf2CT. The measured value is 5.8 μ M, which is 3.6 times weaker than Taf14ET-Taf2CT interaction. However, authors' statement in this case is: "MST

measurements revealed that Taf14YEATS interacts with Taf2CT only slightly weaker compared to the interaction of Taf14ET with Taf2CT" (lines 100-102). Therefore, the authors' interpretation of the data is inconsistent throughout the manuscript and it is biased towards supporting their hypothesis.

2. In Figure 2G, the authors present a model how Taf2CT may interact simultaneously with Taf14 YEATS and ET domains. This model is based on the claim that "Evaluation of the TFIIF complex, and particularly Taf14FL, in the cryo-EM structure of the PIC (Robinson et al., 2016, Cell 166:1411) showed that Taf14YEATS and Taf14ET are in close proximity...". In this paper, the structure of Mediator-RNAPII-PIC was studied and Taf14 protein was present as a TFIIF subunit (Tfg3). Although this proposed structure of Taf14 may be correct in context of RNAPII and TFIIF, it does not necessarily apply to TFIID (structure of which was not determined in the Robinson et al. paper). Presenting this model (Fig. 2G) as a solid result of the study is too speculative, especially in the light of the data presented in the Figure 2H.

3. Authors suggest that Taf14 YEATS and ET domains are in the close proximity and the Taf2CT peptide can bind both domains simultaneously. To support this idea, the DSSO cross-linking assay was performed and the data is presented in the Figure 2H. The result shows clearly that Taf2CT cross-links very efficiently to the linker and YEATS domains of Taf14, but not to ET domain. Only one weak interaction with ET was detected in the region of Taf14 amino acid residue 240, which does not co-localize with peptide binding site shown in Figures 1E and 1F. However, the authors ignore the data and state that "... Taf2CT was found to be cross-linked to both domains and to the linker between Taf14ET and Taf14YEATS" (lines 116-117). The actual Taf2CT cross-link sites should be presented on the 3D model of Taf14 to show that these sites are in proximity indeed. Data from the Figure 2H suggests that these sites are around Taf14 amino acid residues 10, 50, 80, 100, and 125. Figure 2G suggests that the YEATS-domain binding site of the peptide in Taf14 is located around the amino acid 100. Where are other cross-link sites on that 3D structure?

4. Figure 2i and its interpretation are misleading because incomparable results are compared. Mutated Taf2CT peptide was used to prove the specificity of the binding and the result is presented in Figure 2i. First two panels are supposed to show that mutated peptide does not bind to the YEATS domain of Taf14. In these experiments the YEATS:peptide ratio was 1:2, while in results with wt peptide (Fig. 2A) the ratios were 1:4 and 1:6. To make a statement that wt peptide binds to YEATS and mutated peptide does not, these experiments should be performed in the same conditions.

5. The model presented in Figure 3M is not consistent with the actual data presented in the Figure 3J. Figure 3J shows that upon addition of DNA, the cross-links between YEATS and the rest of the protein are lost. However, ET and linker do still crosslink and they do it exactly in the same way as they did in the absence of DNA (Fig. 3i). Therefore, the data does not support the model presented in the last panel of the Figure 3M.

6. It is not clear how was done the experiment shown in Figure 3L. If authors intend to show that DNA binding of Taf14 does not affect its ability to interact with H3K9cr, then the binding experiments with and without DNA should be presented side by side in comparable experimental conditions.

7. Taf14 binding to nucleosomal DNA (Fig 4a-c). The control experiment with Taf14-W81A (or Taf14-delta YEATS) together with Taf2CT peptide is missing. Only Taf14-W81A experiment without the peptide is shown in Suppl Fig. 2C. It is essential to show that wt, W81A and deltaYEATS proteins respond to the Taf2CT peptide in identical fashion proving that the YEATS-mediated interactions with the nucleosomal template are not responsible

for the band-shifts. Also, the reaction with Taf2CT peptide alone should be presented to exclude its own binding to the template (especially taking into account that the peptide was added into these assays in ten-fold excess compared to Taf14 proteins).

8. The model in the Figure 4E is too speculative and is not supported by the actual data. There are two major problems: i) the cross-linking data do not support the view that Taf2CT interacts with Taf14ET, and ii) no evidence has provided that a single Taf2CT peptide can bind both domains simultaneously. In all experiments, the Taf2CT has been present in excess compared to the Taf14 and therefore, several peptides can bind to the same Taf14 molecule. To increase plausibility of the proposed model, the authors should provide direct evidence that a single Taf2CT peptide is capable to interact with both domains simultaneously.

9. The results shown in Figure 4F (RT-qPCR) do not support any conclusions allegedly made on the basis of this experiment. First, one of the genes referred as "not regulated by Taf14" (line 179) does not stand out in the figure – YAL017W looks indistinguishable from YPR145W and YPR158W. In which respect the latter two are "Taf14 responsible" and the first is "not regulated by Taf14"? Second, the actual data shows that wt, 7K/Rmut and W81A alleles have essentially identical effects on the expression of all genes presented in this figure. However, completely ignoring this result, the authors state that "... additionally, the Taf147K/Rmut linker mutant showed partial loss of Taf14-mediated repression of several genes tested. In contrast, these genes were only weakly impacted by the loss of the H3K9acyl binding activity of Taf14" (lines 185-187). This statement is totally wrong. In addition, the authors comment the results of the Taf14-deltaET mutant: "... the deletion of the ET domain resulted in the significant loss of Taf14-mediated suppression of many genes tested ..." (lines 183-184). In fact, if we define the possible range of gene expression in the scale where "wt Taf14 cells" represent the full Taf14 activity and "empty vector cells" define the loss of Taf14 function, then the ET mutant is in the most cases "wt"-like, not "empty vector"-like.

In addition, this experiment must be described in more detail. Where did "the set of Taf14 responsible genes" come from? Is this selection based on some published results?

10. To prove the relevance of Taf14 DNA binding *in vivo*, the efficiency of recruitment of wt and 7K/R Taf14 proteins to their response promoters should be compared by the ChIP assay.

11. The results shown in Figure 4G do not present any new data. The fact that ET-deficient Taf14 is unable to complement taf14-delta cells, is well known and already published several times (for example in Schulze et al., 2010, Mol. Genet. Genomics 283:365). Although the authors present this data to support their conclusions on the importance of Taf2-Taf14 interaction, the main challenge of interpretation of these results is the fact that Taf14 is a component of several important protein complexes (TFIIF, TFIID, NuA4, SWI/SNF, RSC, INO80) and therefore, it is very hard to connect the mutant Taf14 phenotypes to any particular complex. To make a clear connection to TFIID, authors should show that upon disruption of Taf2-Taf14 interaction, the expression of wt Taf14 does not rescue the taf14-delta phenotype. This can be done for instance in the strain where Taf2 is lacking the last 15 amino acids – i.e. the Taf2CT peptide sequence is deleted.

12. Lines 200-201. Authors state that Taf2 amino acids 775-780 are required for Taf2-DNA interaction and refer to the Feigerle & Weil 2016 (JBC, 291:22721). DNA interaction properties of this mutant (m32) were not tested in this paper. The possible role of these residues in DNA interaction were proposed on basis of homology of yeast Taf2 with the human protein and the structure of the latter was determined by Louder et al. 2016 (Nature, 531:604). However, according to Feigerle & Weil 2016, the yeast and human Taf2

proteins show only 32% similarity, making it very speculative to claim that yeast Taf2 residues 777-780 are actually involved in DNA binding. Authors should provide the experimental evidence that this mutant is deficient in DNA binding before presenting that statement as a solid fact and building their story on that.

Minor points:

1. Lines 54-55. Authors state: "Genetic studies have shown that the interaction between Taf14 and another subunit of TFIID complex, Taf2, is essential in the TFIID assembly and function (Feigerle & Weil 2016)". However, in the referred paper the authors show clearly that TFIID complex forms in the absence of Taf14 (or the Taf2 C-terminal domain) in the same way as in wt cells and the Taf2-deltaC cells grow only slightly slower than their wt counterparts. Therefore, "... is essential in the TFIID assembly and function ..." is clear exaggeration of the real situation.

2. In line 211 the authors state that "... our findings demonstrate that Taf14 exists in an autoinhibited state ...". This gives an impression that there is free Taf14 in the cells that waits for activation. Similar misleading statement is also present in the Abstract of the manuscript (lines 33-34). So far, Taf14 has been found only in protein complexes and there is no evidence of free Taf14 in the cells. Authors do not provide any evidence that Taf14 can exist in different conformations when incorporated into TFIID complex, therefore the statement that Taf2 turns Taf14 into active form is rather artificial. Basically, a similar statement can be made for any multi-subunit protein complex where the "active" conformation of single proteins is achieved by formation of the complex.

3. Line 52. Refer to Fig. 1A (not just Fig. 1).

4. Line 170. Reference to the wrong figure. Should be Suppl. Fig. 2d and 2e, not 1d and 1e.

5. Line 202. Reference to the wrong figure. Should be Fig. 4H, not 4G.

6. Apparently, in this manuscript the term "RT-qPCR" stands for "reverse-transcriptase quantitative PCR" as the mRNA levels of selected genes were determined. In the text it has been referred as "real-time quantitative PCR". Please correct (lines 177-178 and 416).

7. It is not clear which fragment of Taf14 was expressed as Taf14ET. In the line 238 it is written: "... Taf14ET (aa 162-243, aa 168-243) constructs...". What does it mean? Is it 162-243, or 168-243, or a mixture of both? Please clarify.

8. The cell dilution ratio in spotting assays (Figs. 4G and 4H) is not indicated. Visual estimation of colony numbers in the last spots suggests that this is not a traditional ten-fold dilution series. It looks more like 3-5 fold dilutions. Please clarify the assay conditions.

9. Show the full gel pictures of the EMSA assays, including the loading wells in Figures 3 and 4.

10. Add a gel picture of all purified proteins into the supplementary material.

Reviewer #3 (Remarks to the Author):

In this manuscript, the authors utilized an elegant biochemical approach to determine

Taf14 domains that binds to the C-terminal tail of Taf2. These interactions further elicit a domain rearrangement in Taf14, which release the linker region of Taf14 to engage with DNA and the nucleosome. Deletion mutants that interrupt Taf14/Taf2 interactions displayed transcription dysregulation in target genes, providing strong validation for the biochemical study. Overall, the structural and biochemical characterization of taf14 and Taf2 interactions seems thorough with multiple approaches and the scope suitable for Nature Communications. However, this reviewer has a major concern and some minor suggestions for authors to consider and ensure scientific rigor:

Major concern: authors seem to consider free DNA and nucleosome equivalently in result presentation and discussion (Fig. 4e). However, their data speak differently. For instance, Taf14FL binds to free 601 DNA well (Fig. 3b), but not nucleosome (Fig. 4a). W81A mutation disrupts nucleosome interaction (Sup Fig. 2c), but not free 601 DNA interaction (Sup Fig. 2a). Although authors established importance of Taf14 linker region to interact with free DNA in Fig. 3c, the role of Taf14 linker region in nucleosome binding was not shown. Therefore, the conclusion in Fig. 4e is not fully supported.

Minor suggestions:

- 1. Presentation of cross-linked pairs in Fig. 2h, 3i and 3j should be mapped to 3D models for readers to judge quality of data. Similarly, representative MS/MS spectra of cross-linked peptides should be shown for readers to assess data quality.**
- 2. In line 130-135, specific supplement figures in Sup Fig. 2 should be pointed out in data presentation to help readers navigate through the supplemental figures.**
- 3. In line 168-170, there is no Sup Fig 1d and 1e. The whole sentence was not supported by data.**

We thank the Editor and Reviewers for the insightful and very constructive comments, which were helpful in revising and strengthening this manuscript.

New data are shown in Figs.: 2i, two left panels, 4c, 4g and in Suppl. Figs. 4a, 4b, 5b, 5c, 5d, 5e, 10b, 10c, 10d, 10e, 10f, 10g, 10h, 10i, 11

New analysis is shown in Fig. 4f and Suppl. Figs.: 3, 6a, 6b

New figures are added: Suppl. Fig. 2

Reviewer 1, Comment 1: Together, this study presents an impressive amount of work identifying the molecular mechanism of the Taf14-Taf2 interaction and its possible role in modulating the Taf14-NCP interaction... The identity of Taf2CT is never defined in the main text... How do authors determine the minimum fragment of Taf2CT required for Taf14ET interaction? If cited from a previous publication, the appropriate reference is needed.

Author's response: we cite ref. 13: "...the ET domain is capable of associating with the hxxK motif found in several co-activators, such as Sth1 and Taf2¹³." and "... examined their association with the Taf2 peptide corresponding to the C-terminal hxxK motif-containing tail of Taf2..."

Reviewer 1, Comment 2: In the 2016 JBC paper, GST pull-down assay showed Taf14YEATS did not bind Taf2. Thus, this discrepancy needs to be explained – the conditions in a single pull-down assay in 2016 JBC paper were not varied and may not be optimal to detect binding. Our NMR and MST experiments clearly show the interaction.

How Taf14ET binds Taf2CT remains unsolved – the structure of the Taf14ET-Taf2CT complex is shown in Fig1.

First, according to the structural model (Fig. 2g), Taf2CT must anchor to Taf14ET, and then the N-terminal 2-3 residues of Taf14CT make contacts with Taf14YEATS. It is very unlikely that Taf14YEATS (in the absence of Taf14ET) binds Taf2CT using the same contacting interface. Such a small contacting Taf14YEATS-Taf2CT interface, as seen in Fig. 2g, cannot support the strong binding between Taf14YEATS and Taf2CT (Kd: 6 μ M). – we have added Suppl. Fig. 2 to clarify the orientation of the Taf2 peptide with respect to the YEATS domain. It is really fascinating and seems the Taf2 peptide pairs with the beta-strand of the YEATS domain in an antiparallel manner. We don't elaborate such a pairing to avoid overstatement. But I am positive that in the cryo-EM structure, a tail of another protein is sandwiched between the YEATS and ET domains (TK). Obviously, whatever is bound between these domains forming a contiguous twisted beta sheet cannot be observed at this resolution of the cryo-EM structure.

Both beta-strands of the YEATS domain (at one of the edges of the beta sandwich) are perturbed in NMR experiments, suggesting that Taf2 might form the additional beta-strand at this edge (also corroborating the overall orientation of the YEATS-(ligand)-ET assembly in the cryo-EM structure), bridging the existing beta-sheets of the YEATS and ET domains.

Of note, a few YEATS domain resonances that are perturbed cannot be assigned (the assignments are taken from ref. 26), therefore the binding interface can be more extended.

Second, Taf2 V1397D and I1399D completely disrupt the interaction between Taf14YEATS and Taf2CT, but this is not supported by the proposed structural model in which Taf2 V1397 and I1399 make no contact with Taf14YEATS. – we don't have the structure of the YEATS-Taf2 complex, therefore we cannot speculate how V1397 and I1399 of Taf2 contribute to the interaction. One possibility is that Val and Ile (which are well-known to prefer to be in beta-strands, especially in the middle of the strands) are parts of a beta-strand. It is also well-known that Asp favors alpha-helices.

Replacement of Val or Ile with Asp should decrease or disrupt the beta-sheet contacts with the beta-sandwich of the YEATS domain.

Third, Taf2 bridges two Taf14 domains together to favor a closed conformation but not an “open” conformation in this structural model. It contradicts the authors’ discussion (line 213-215) “Binding of Taf2 to the ET domain and the YEATS domain of Taf14 eliminates the autoinhibition: it causes a substantial rearrangement of Taf14, resulting in the release of the linker region which binds to DNA via the positively charged motifs”. – there is no contradiction, because ‘autoinhibition’ is referred to the association with the internal linker region in the apo state and the inability of the full-length protein to interact with the nucleosome. When the linker is not available (not exposed), Taf14FL weakly binds to DNA.

Reviewer 1, Comment 3: ...Can Taf14ET or Taf14YEATS alone bind DNA? Can Taf14Linker bind DNA without both Taf14ET and Taf14YEATS? Is there any DNA sequence preference or specificity for Taf14?... It is also better to quantify the binding affinity (or K_d) of the Taf14-DNA interaction to show how significant this interaction is.

Author’s response: suggested EMSA experiments with Taf14ET and Taf14YEATS (new data are shown in Suppl. Figs. 4a, b) indicate that the isolated domains do not appreciably bind DNA, however, we were unable to express a soluble ¹⁵N-labeled linker construct to perform NMR experiments with DNA. The following sentence has been added on page 6: “We found that Taf14_{YEATS} does not bind to 601 DNA, supporting previous findings ¹⁵, and Taf14_{ET} shows a very weak DNA binding activity (Suppl. Fig. 4a, b).”

Taf14 has a weak selectivity toward AT-rich DNA as compared to GC-rich DNA (new data are shown in Suppl. Figs. 5b-e and discussed on page 7).

Since 601 DNA has ~13 major/minor grooves, it’s difficult to calculate K_ds toward DNA without knowing the stoichiometry of binding.

Reviewer 1, Comment 4: Although I am not an expert on SAXS, I still have several concerns about SAXS data. First, the data quality shown by the Guinier plot (Supplementary Fig. 4c) is low, indicating the conclusion from the apo Taf14FL is not reliable.

Author’s response: we performed measurements at very low concentrations to avoid the concentration effect, hence, signal to noise ratio is decreased, however, the most important feature which defines quality is the consistency of the linearity of the plots at all concentrations (please see figure on the left). Moreover, for the apo Taf14FL we analyzed data in the range up to 5 nm⁻¹, not just the q range in the Guinier region.

Second, the Kratky plots of apo Taf14FL and Taf14FL+Taf2CT indicate a partially unfolded structure,

not just “flexible” or “several conformations in equilibrium.” - it’s not unfolding- both domains are folded based on HSQC spectra, see figures on the left.

Furthermore, we have further analyzed the Taf14FL SAXS data using the ensemble modelling method. The 10,000 conformations were used to fit to the data, and the ensemble of three conformations was found to best fit to the SAXS data (see figures below; YEAST domains from all three conformations are aligned and colored magenta).

Third, the SAXS data is not fully consistent with the role of Taf2CT in inducing a conformational change of Taf14. Authors proposed that Taf2CT binding Taf14 could induce Taf14 rearrangement to form the “open” conformation to enable its interaction with DNA, but the SAXS data showed that Taf14FL and Taf14FL+Taf2CT have similar SAXS curves – this is incorrect, we do not propose that “Taf2CT binding to Taf14 could induce Taf14 rearrangement to form the “open” conformation”.

As shown in Fig. 4e, Taf2 is sandwiched between the YEATS and ET domains (it’s not an open state) resulting in the release of the internal region – the linker – from the YEATS/ET for binding to DNA. We should expect similarity between the apo-state of Taf14 and Taf14FL+Taf2CT because the linker plays a role somewhat similar to Taf2 in the absence of Taf2 (shown in Fig. 3l, left panel and Fig. 4e, right panel).

Reviewer 1, Comment 5: ... Taf14ΔYEATS, as well as Taf14YEATS and Taf14ET alone, should be tested by the similar EMSA as Taf14ΔET (Fig.4b) to see which Taf14 domain is important for NCP binding. – as suggested, we have performed EMSAs with Taf14ΔYEATS, Taf14YEATS and Taf14ET (new data are shown in Suppl. Figs. 10b-d).

Second, WT NCP should be tested to check whether the Cr modification plays a dominant role in the Taf14-NCP interaction or the binding is dominantly determined by the Taf14-DNA interaction. – WT NCP is now tested (new data are shown in Suppl. Fig. 10h).

Third, considering Taf14ET could bind a lot of short motifs conserved as Taf2CT, will other Taf14-binding peptides, e.g., Sth1 peptide, have the similar effect of promoting the Taf14-H3K9cr NCP binding? - to avoid being unfocused, in this study we do not test other reported ligands of Taf14ET.

Reviewer 1, Comment 6: Fig. 4f: First, the control genes YLR290 and YAL017 should be included in the figure. Second, the significance of the differences between bars should be shown. Third, I don’t see much difference between Taf14 W81A and 7K/R mutant (only show the difference on two genes out of 12 tested genes). – we have redone this figure and revised the text to make sure the results are more

easily understandable. The rescue of WT full-length *TAF14* brings expression back to normal or at 1, which is our comparison for the other mutants and vector only that represents the *TAF14* deletion. We have also included statistical analysis and added more detail on the experiments.

More importantly, to ascertain that association of Taf14 with Taf2 is essential for Taf14-mediated transcriptional regulation, the Taf14 or Taf2 mutants (point mutations instead of truncations) disrupting the Taf14-Taf2 interaction should be examined...- we have tried extensively to make point mutants in Taf2 without success however, the C-terminus truncation mutant of Taf2 worked well.

Some typos:

Line 170: should be Suppl. Figs. 2d and 2e and Line 202: should be Fig.4h - corrected, thank you

Reviewer 2, Comment 1: In Figure 1C, the authors determine K_d of Taf14-ET binding to Taf2CT. The value is 1.6 and when comparing it to the full-length Taf14-Taf2CT binding (Fig. 1H; $K_d=0.7$ μ M) the authors state that "... binding of full-length protein is two-fold tighter..." (line 93) emphasizing the importance of this difference. In figure 2B the authors measure the K_d of Taf14-YEATS interaction with Taf2CT. The measured value is 5.8 μ M, which is 3.6 times weaker than Taf14ET-Taf2CT interaction. However, authors' statement in this case is: "MST measurements revealed that Taf14YEATS interacts with Taf2CT only slightly weaker compared to the interaction of Taf14ET with Taf2CT" (lines 100-102).

Author's response: this sentence has been revised to: "MST measurements revealed that Taf14_{YEATS} interacts with Taf2_{CT} with a K_d of 6 μ M..."

Reviewer 2, Comment 2: In Figure 2G, the authors present a model how Taf2CT may interact simultaneously with Taf14 YEATS and ET domains. This model is based on the claim that "Evaluation of the TFIIF complex, and particularly Taf14FL, in the cryo-EM structure of the PIC (Robinson et al., 2016, Cell 166:1411) showed that Taf14YEATS and Taf14ET are in close proximity...". In this paper, the structure of Mediator-RNAPII-PIC was studied and Taf14 protein was present as a TFIIF subunit (Tfg3). Although this proposed structure of Taf14 may be correct in context of RNAPII and TFIIF, it does not necessarily apply to TFIID (structure of which was not determined in the Robinson et al. paper). Presenting this model (Fig. 2G) as a solid result of the study is too speculative, especially in the light of the data presented in the Figure 2H.

Author's response: we clearly state that it is TFIIF complex in the text and Figure 2g legend and cite ref. 4. We never state that this is the current result. We have added: "... in the previously reported cryo-EM structure of the PIC (ref. 4)" on page 5. Of note, Tfg1 might play a role of Taf2 in the TFIIF complex, being sandwiched between the YEATS and ET domains (confidential please).

Reviewer 2, Comment 3: [MS data, Fig. 2h]... The result shows clearly that Taf2CT cross-links very efficiently to the linker and YEATS domains of Taf14, but not to ET domain. Only one weak interaction with ET was detected in the region of Taf14 amino acid residue 240, which does not co-localize with peptide binding site shown in Figures 1E and 1F. However, the authors ignore the data and state that "... Taf2CT was found to be cross-linked to both domains and to the linker between Taf14ET and Taf14YEATS" (lines 116-117). The actual Taf2CT cross-link sites should be presented on the 3D model of Taf14 to show that these sites are in proximity indeed. Data from the Figure 2H suggests that these sites are around Taf14 amino acid residues 10, 50, 80, 100, and 125. Figure 2G suggests that the YEATS-domain binding site of the peptide in Taf14 is located around the amino acid 100. Where are other cross-link sites on that 3D structure?

Author's response: as suggested, the cross-link sites have been mapped on the structure in Suppl. Fig. 3, and the following sentences have been revised/added: "The possibility of such an engagement was corroborated by mass spectrometry analysis of DSSO cross-linked Taf14_{FL} in the presence of Taf2_{CT} (Fig. 2h, Suppl. Fig. 3 and Suppl. Table 2). In the Taf14_{ET}-Taf2_{CT} complex, the only lysine residue of Taf2_{CT}, K1398, is in the middle of the β -strand and thus its position is fixed. The side chain of K1398 is perpendicular to the surface of Taf14_{ET}, pointing straight away from the domain (Suppl. Fig. 3), and cannot be easily crosslinked backward to Taf14_{ET}, as only a single weak contact was detected by MS (Fig. 2h). However, K1398 is readily crosslinked with several lysine residues of Taf14_{YEATS}, located in the same part of Taf14_{YEATS} that was perturbed in NMR experiments (Fig. 2h and Suppl. Fig. 3).

Reviewer 2, Comment 4: Figure 2i and its interpretation are misleading because incomparable results are compared...In these experiments the YEATS:peptide ratio was 1:2, while in results with wt peptide (Fig. 2A) the ratios were 1:4 and 1:6. To make a statement that wt peptide binds to YEATS and mutated peptide does not, these experiments should be performed in the same conditions.

Author's response: the experiments were re-run using ratios 1:4 and 1:6. New data are shown in Fig. 2i, two left panels.

Reviewer 2, Comment 5: ... Figure 3J shows that upon addition of DNA, the cross-links between YEATS and the rest of the protein are lost. However, ET and linker do still crosslink and they do it exactly in the same way as they did in the absence of DNA (Fig. 3i). Therefore, the data does not support the model presented in the last panel of the Figure 3M.

Author's response: the last panel of the Fig. 3I (former 3m) has been removed.

Reviewer 2, Comment 6: It is not clear how was done the experiment shown in Figure 3L. - we have previously shown that the isolated YEATS domain binds to H3K9acyl; in Fig. 3I (now Fig. 3m) we demonstrate that Taf14_{FL} binds to H3K9cr and the H3K9cr-binding site is not occluded in the full-length protein.

Reviewer 2, Comment 7: Taf14 binding to nucleosomal DNA (Fig 4a-c). The control experiment with Taf14-W81A (or Taf14-delta YEATS) together with Taf2_{CT} peptide is missing. Only Taf14-W81A experiment without the peptide is shown in Suppl Fig. 2C. It is essential to show that wt, W81A and deltaYEATS proteins respond to the Taf2_{CT} peptide in identical fashion proving that the YEATS-mediated interactions with the nucleosomal template are not responsible for the band-shifts. Also, the reaction with Taf2_{CT} peptide alone should be presented to exclude its own binding to the template (especially taking into account that the peptide was added into these assays in ten-fold excess compared to Taf14 proteins).

Author's response: as suggested, we performed EMSA experiments with NCPs using Taf14-W81A + Taf2_{CT} peptide, Taf14-delta YEATS + Taf2_{CT} peptide, and separately Taf2_{CT} peptide (new data are shown in Suppl. Figs. 10e, i). We also re-ran EMSA to have only 2-fold excess of Taf2 (in new Fig. 4c) and performed additional EMSAs again using 2-fold excess of Taf2 (shown in Suppl. Figs. 5d, e, and 10e, g, h).

Reviewer 2, Comment 8: The model in the Figure 4E is too speculative and is not supported by the actual data. There are two major problems: i) the cross-linking data do not support the view that Taf2_{CT} interacts with Taf14_{ET} – we respectively disagree, the crystal structure of the complex shown in Fig. 1,

NMR experiments and measured binding affinities show that Taf2CT interacts with Taf14ET. For MS analysis, please see our response to Comment 3.

, and ii) no evidence has provided that a single Taf2CT peptide can bind both domains. In all experiments, the Taf2CT has been present in excess compared to the Taf14 and therefore, several peptides can bind to the same Taf14 molecule. To increase plausibility of the proposed model, the authors should provide direct evidence that a single Taf2CT peptide is capable to interact with both domains simultaneously. – the experiments were re-run with the ratio of 1:2 (Taf14 constructs :Taf2 peptide), and the conclusions remain the same - new data are shown in Fig. 4c and Suppl. Figs. 10e, g, h, i.

We have set up numerous co-crystallization trays but haven't obtained diffracting crystals of the complex yet, this work is in progress. We are also working on the cryo-EM structure of the complex.

Reviewer 2, Comment 9: The results shown in Figure 4F (RT-qPCR) do not support any conclusions allegedly made on the basis of this experiment. First, one of the genes referred as “not regulated by Taf14” (line 179) does not stand out in the figure – YAL017W looks indistinguishable from YPR145W and YPR158W. In which respect the latter two are “Taf14 responsible” and the first is “not regulated by Taf14”? Second, the actual data shows that wt, 7K/Rmut and W81A alleles have essentially identical effects on the expression of all genes presented in this figure. However, completely ignoring this result, the authors state that “... additionally, the Taf147K/Rmut linker mutant showed partial loss of Taf14-mediated repression of several genes tested. In contrast, these genes were only weakly impacted by the loss of the H3K9acyl binding activity of Taf14” (lines 185-187). This statement is totally wrong. In addition, the authors comment the results of the Taf14-deltaET mutant: “... the deletion of the ET domain resulted in the significant loss of Taf14-mediated suppression of many genes tested ...” (lines 183-184). In fact, if we define the possible range of gene expression in the scale where “wt Taf14 cells” represent the full Taf14 activity and “empty vector cells” define the loss of Taf14 function, then the ET mutant is in the most cases “wt”-like, not “empty vector”-like. In addition, this experiment must be described in more detail. Where did “the set of Taf14 responsible genes” come from? Is this selection based on some published results?

Author's response: we have revised the text and redone this figure to make sure the results are more easily understandable. The rescue of WT full-length *TAF14* brings expression back to normal or at 1, which is our comparison for the other mutants and vector only that represents the *TAF14* deletion. We have also included statistical analysis and added more detail on the experiments.

Reviewer 2, Comment 10: To prove the relevance of Taf14 DNA binding in vivo, the efficiency of recruitment of wt and 7K/R Taf14 proteins to their response promoters should be compared by the ChIP assay.

Author's response: we thank the reviewer for this suggestion and have added ChIP data (Fig. 4g). It is indeed very helpful, as it shows the substantial impact the ET domain loss and the linker mutant has on Taf14 recruitment to genes.

Reviewer 2, Comment 11: The results shown in Figure 4G do not present any new data. The fact that ET-deficient Taf14 is unable to complement taf14-delta cells, is well known and already published several times (for example in Schulze et al., 2010, Mol. Genet. Genomics 283:365). Although the authors present this data to support their conclusions on the importance of Taf2-Taf14 interaction, the main challenge of interpretation of these results is the fact that Taf14 is a component of several important protein complexes (TFIIF, TFIID, NuA4, SWI/SNF, RSC, INO80) and therefore, it is very hard to connect the mutant Taf14 phenotypes to any particular complex. To make a clear connection to

TFIID, authors should show that upon disruption of Taf2-Taf14 interaction, the expression of wt Taf14 does not rescue the taf14-delta phenotype. This can be done for instance in the strain where Taf2 is lacking the last 15 amino acids – i.e. the Taf2CT peptide sequence is deleted.

Author's response: in these experiments (now Fig. 4h) we compare the effect of delta-ET, the mutation of YEATS and mutations in the linker, which is a new, unpublished data. We agree, because Taf14 is a component of several important protein complexes, the only conclusion we could draw is the importance of the ET domain in overall multifaceted Taf14 function (second paragraph on page 9). We show the effect of delta C-terminus of Taf2 in Fig. 4i. We also cite Schulze et al., 2010 (ref. 17).

Reviewer 2, Comment 12: Lines 200-201. Authors state that Taf2 amino acids 775-780 are required for Taf2-DNA interaction and refer to the Feigerle & Weil 2016 (JBC, 291:22721). DNA interaction properties of this mutant (m32) were not tested in this paper. The possible role of these residues in DNA interaction were proposed on basis of homology of yeast Taf2 with the human protein and the structure of the latter was determined by Louder et al. 2016 (Nature, 531:604). However, according to Feigerle & Weil 2016, the yeast and human Taf2 proteins show only 32% similarity, making it very speculative to claim that yeast Taf2 residues 777-780 are actually involved in DNA binding. Authors should provide the experimental evidence that this mutant is deficient in DNA binding before presenting that statement as a solid fact and building their story on that.

Author's response: we do not further investigate the m32 construct in this manuscript and refer to the previous studies (ref. 14), but we do observe a more notable effect in Taf2mut_deltaC, and therefore cannot ignore the possibility of a more complex mechanism for the association of Taf2 with Taf14/DNA/chromatin. We have added that the functional importance of this region needs to be examined in future studies (page 11).

Minor points:

1. Lines 54-55. Authors state: “Genetic studies have shown that the interaction between Taf14 and another subunit of TFIID complex, Taf2, is essential in the TFIID assembly and function(Feigerle & Weil 2016)”. However, in the referred paper the authors show clearly that TFIID complex forms in the absence of Taf14 (or the Taf2 C-terminal domain) in the same way as in wt cells and the Taf2-deltaC cells grow only slightly slower than their wt counterparts. Therefore, “... is essential in the TFIID assembly and function ...” is clear exaggeration of the real situation.

Author's response: The conclusion of the Feigerle & Weil 2016 JBC paper is that “mutation in Taf2 can completely disrupt the ability of Taf14 to associate with the TFIID complex. Taf14-less TFIID-containing cells display defects in growth and transcript abundance for the highly transcribed TFIID-dominated ribosomal protein-encoding genes.... data indicate that the Taf14 YEATS domain contributes to TFIID function.”

2. In line 211 the authors state that “... our findings demonstrate that Taf14 exists in an autoinhibited state ...”. This gives an impression that there is free Taf14 in the cells that waits for activation. Similar misleading statement is also present in the Abstract of the manuscript (lines 33-34). So far, Taf14 has been found only in protein complexes and there is no evidence of free Taf14 in the cells. Authors do not provide any evidence that Taf14 can exist in different conformations when incorporated into TFIID complex, therefore the statement that Taf2 turns Taf14 into active form is rather artificial. Basically, a similar statement can be made for any multi-subunit protein complex where the “active” conformation of single proteins is achieved by formation of the complex. – the Taf14 synthesis obviously results in a free protein, not the complex. Furthermore, there is no evidence that free Taf14 cannot be present in

the nucleus or while shuttling between the complexes.

3. Line 52. Refer to Fig. 1A (not just Fig. 1). - done

4. Line 170. Reference to the wrong figure. Should be Suppl. Fig. 2d and 2e, not 1d and 1e. - done

5. Line 202. Reference to the wrong figure. Should be Fig. 4H, not 4G. - done

6. Apparently, in this manuscript the term “RT-qPCR” stands for “reverse-transcriptase quantitative PCR” as the mRNA levels of selected genes were determined. In the text it has been referred as “real-time quantitative PCR”. Please correct (lines 177-178 and 416). – corrected, thank you for catching

7. It is not clear which fragment of Taf14 was expressed as Taf14ET. In the line 238 it is written: “... Taf14ET (aa 162-243, aa 168-243) constructs...”. What does it mean? Is it 162-243, or 168-243, or a mixture of both? Please clarify. – this has been clarified in methods on pages 11 and 14.

8. The cell dilution ratio in spotting assays (Figs. 4G and 4H) is not indicated. Visual estimation of colony numbers in the last spots suggests that this is not a traditional ten-fold dilution series. It looks more like 3-5 fold dilutions. Please clarify the assay conditions. – dilutions (1:5 in Fig. 4h and 1:4 in Fig. 4i) are stated in method sections on pages 18 and 20.

9. Show the full gel pictures of the EMSA assays, including the loading wells in Figures 3 and 4. – included in the Source File.

10. Add a gel picture of all purified proteins into the supplementary material. – included in the Source file.

Reviewer 3, Comment 1: Major concern: authors seem to consider free DNA and nucleosome equivalently in result presentation and discussion (Fig. 4e). – the model in Fig. 4e is correct- as shown in Figs. 4c, d and Suppl. Figs. 5b-e and 10e, g, h – binding of Taf2 increases binding of Taf14 to DNA and to the nucleosome.

However, their data speak differently. For instance, Taf14FL binds to free 601 DNA well (Fig. 3b), but not nucleosome (Fig. 4a). W81A mutation disrupts nucleosome interaction (Sup Fig. 2c) – this is incorrect, both WT Taf14 and W81A mutant of Taf14 are incapable of binding to the nucleosome (Fig. 4a and Suppl. Fig. 10a). Taf2 is needed to bind to the nucleosome. but not free 601 DNA interaction (Sup Fig. 2a) - both WT Taf14 and W81A mutant of Taf14 bind to DNA equally (Fig. 3b and Suppl. Fig. 4c). Although authors established importance of Taf14 linker region to interact with free DNA in Fig. 3c, the role of Taf14 linker region in nucleosome binding was not shown. – we have performed EMSAs with Taf14 4K/Rmut + NCP (Suppl. Figs. 10f, g).

Minor suggestions:

1. Presentation of cross-linked pairs in Fig. 2h, 3i and 3j should be mapped to 3D models for readers to judge quality of data. Similarly, representative MS/MS spectra of cross-linked peptides should be shown for readers to assess data quality. – as suggested, the MS cross-link sites mapped on the structure are now shown in Suppl. Figs. 3 and 6.

2. In line 130-135, specific supplement figures in Sup Fig. 2 should be pointed out in data presentation to help readers navigate through the supplemental figures. In line 168-170, there is no Sup Fig 1d and 1e... - corrected

REVIEWER COMMENTS

Reviewer #1 (Remarks to the Author):

The revised manuscript has been improved compared to the previous version. Most of my initial concerns were addressed. However, two major concerns are not adequately addressed in the revised manuscript.

1. Both Taf14ET and Taf14YEATS bind to Taf2CT

I don't question that both Taf14ET and TafYEATS binds to Taf2CT, but I don't believe the structural model of one Taf2CT bound both Taf14ET and Taf14YEATS (Fig. 2g). In the revised manuscript, the authors proposed that Taf2CT forms a contiguous twisted beta-strand to pair with both Taf14 YEATS and ET beta-sheets, as shown in Supplementary Fig. 2. There are several problems with this model. First, this model is built from the Taf14 structure from a low-resolution cryo-EM structure of the mediator-bound Pol II structure (only 15Å resolution). Such a low-resolution map cannot provide a precise molecular model. In addition, Taf14 associates with three proteins in this Pol II structure, so this Taf14 structure may not represent the conformation of apo Taf14 or Taf14-Taf2CT.

Second, the structural model shown in Fig. 2g fails to explain why V1397D and I1399D equally disrupted Taf2 interaction with Taf14YEATS and Taf14ET (Fig. 2i). It is very unlikely that Taf2 V1397 and I1399 contact with Taf14YEATS and Taf14ET simultaneously. Even if the authors' structure model is correct, the N-terminal residues of Taf2CT (residues before 1395) pair with Taf14YEATS beta-sheet, but not V1397 and I1399. The authors proposed that V1397D and I1399D mutants abolished the secondary structure of Taf2CT. To dissect how these residues contribute to Taf2CT-Taf14 interaction, other Taf2 mutations (e.g. V1397A or I1399) , which decreased the hydrophobic packing with Taf14 but did not affect the secondary structure, should be tested.

I would suggest another possibility that there are two Taf2CT peptides binding to Taf14ET and Taf14YEATS separately. Both copies of Taf2CT form beta-strand to pair with the existing Taf14ET and Taf14YEATS beta-sheets, respectively, and make hydrophobic contacts with Taf14 through Taf2 V1397 and I1399. Can authors quantify the stoichiometry for Taf2CT binding to Taf14?

2. Association of Taf14 with Taf2 and DNA is essential for transcriptional regulation

In the current manuscript, the authors just showed four genes (their initial submission showed twelve genes) and made conclusions based on these four genes. I think it is cherry-picking. Even for the four genes they showed, two of four genes (ACS1, YKR039W) showed different suppression effects on Taf14 7K/Rmut and Taf14ΔET, so it hardly supported the conclusion "the Taf147K/Rmut linker mutant also showed a loss of Taf14-mediated suppression similar or equal to the loss of Taf14ET" (line 198-199). In addition, in most circumstances (either in the current four genes or the previous twelve genes), there was not much difference between Taf14 7K/R mutant and W81A mutant, so it is not appropriate to make the statement from lines 198-200.

Moreover, authors claimed that "loss of Taf14 acyl-reading had no impact on the ability of Taf14 to associate with these genes". This is not true. As shown in Fig4g, the recruitments of Taf14 to two genes (YER145,YKR039W) genes were greatly increased, and the recruitment to ACS1 was mildly increased. This cannot be concluded as "no impact on the ability of Taf14 to associate with these genes". Why loss of acylation binding activity could increase the recruitment should be discussed. Besides the inconsistency between experimental data and authors' conclusions, some data contradict the previous publication from the same lab, but the authors did not discuss why they had different observations for the same set of cells. For example, in their earlier paper (Shanle et al., Gene & Dev 2015), they claimed that Taf14 W81A grew slower than WT in higher (37°C) or lower temperatures (25°C) (Fig. 4d in Gene & Dev 2015). But in the current manuscript, they stated no difference between Taf14 WT and W81A in the growth rate at different temperatures (line 210-211, Figure 4h).

Reviewer #2 (Remarks to the Author):

The revised manuscript is certainly an improvement of the original version. Technical part of the work is generally much better and it includes appropriate controls in most cases. Nevertheless, there are still several shortcomings, mostly related to the inconsistency of the results and the lack of appropriate discussion.

1. The results from experiments of Taf14 interactions with H3K9cr-NCP need more thorough explanation.

1. 1. The authors have published previously that the Taf14 YEATS domain interacts specifically with H3K9cr (Andrews et al., 2016; Nat. Chem. Biol. 12:396) and the same result is also presented in this manuscript. However, neither full-length Taf14, nor YEATS domain of it can interact with nucleosomes containing H3K9cr. That discrepancy must be explained and appropriately discussed.

1. 2. Some of the presented results are self-contradictory. The authors show that 4K/R mutant is deficient in binding to DNA (Fig. 3C) and to H3K9cr-NCP (Suppl. Fig. 10F). On the other hand, addition of Taf2CT to the binding assay activates 4K/R binding to H3K9cr-NCP (Suppl. Fig. 10G). If none of the components (YEATS domain, 4K/R mutant, Taf2CT) can bind DNA or nucleosomes, then how does Taf2CT activate 4K/R binding to H3K9cr-NCP? This has not explained in the manuscript.

2. Suppl. Fig. 11. Why is Taf14-ET-delta protein not detected with HA antibody? Apparently, the protein is expressed from the pRS313-Taf14(1-132)-HA3-SSN6 plasmid (Suppl. Table 4) and it should contain a C-terminal 3xHA tag. If it is not expressed correctly (as the Suppl. Fig. 11 suggests), then how can authors trust the results with that expression construct (presented in Figs. 4F-H)?

3. Even assuming that Taf14-ET-delta protein is correctly expressed, it lacks the ET domain, which is required for Taf14 interaction with all protein complexes (including TFIID) and the entire linker domain that apparently interacts with DNA. In addition, the linker region overlaps with the nuclear localization signal of Taf14 protein and therefore the Taf14-ET-delta protein is expected to localize in the cytoplasm. How can this protein rescue the growth of taf14-delta cells (Fig. 4H) if it is unable to incorporate into protein complexes, lacks DNA binding ability and is presumably incorrectly localized? These issues should be discussed in the manuscript.

3a. The labelling of Taf14 constructs throughout the manuscript is confusing. It seems that in in vitro experiments (Figs. 1-4B) "Taf14-ET-delta" refers to the Taf14 protein lacking amino acids 172-244 (line 258 in the Methods section). However, in in vivo experiments (Fig. 4F-H), the identically labelled construct lacks aa 133-244 (Suppl. Table 4), which is almost identical to the "Taf14-YEATS" protein (lacking aa 138-244) used in in vitro experiments. To be consistent, the "Taf14-ET-delta" in Figures 4F-H should be labelled as "Taf14-YEATS" (and corrected in the main text as well). Preferably, the aa residue numbers of different Taf14 domains should be indicated also in Figure 1A.

4. The 7K/R mutant of Taf14 appears to be deficient in repression of Taf14-dependent genes (Fig. 4F), while it is fully active in rescuing of taf14-delta cells in the spot test assay (Fig. 4H). Collectively, these results suggest that the DNA binding activity of Taf14 is important for its function (Fig. 4F) and in the same time that it is not important at all (Fig. 4H). This needs a plausible explanation and discussion.

Line 100: reference to Fig. 1C is not appropriate.

Line 138: reference to Fig. 4C is incorrect, it should be 3C.

Line 187: reference to Suppl. Fig. 6H,I is incorrect, presumably it should be Suppl. Fig. 10.

Reviewer #3 (Remarks to the Author):

In this revision, the authors have addressed most of the concerns raised from last review. Additional

results provided in Supplemental Fig 10 e, f, g are especially helpful to support their model illustrated in Fig. 4e. I support the publication of this manuscript.

We thank the Editor and Reviewers for the insightful and very constructive comments, which were helpful in revising and strengthening this manuscript.

Reviewer 1, Comment 1: ... Most of my initial concerns were addressed. However, two major concerns are not adequately addressed in the revised manuscript.

... In the revised manuscript, the authors proposed that Taf2CT forms a contiguous twisted beta-strand to pair with both Taf14 YEATS and ET beta-sheets, as shown in Supplementary Fig. 2. – this is incorrect, we do not propose or state “that Taf2CT forms a contiguous beta-strand to pair with both Taf14 YEATS and ET beta-sheets” anywhere in the manuscript.

There are several problems with this model.

First, this model is built from the Taf14 structure from a low-resolution cryo-EM structure of the mediator-bound Pol II structure (only 15Å resolution). Such a low-resolution map cannot provide a precise molecular model. – we agree, and because of that, we do not speculate how the YEATS domain binds to Taf2. There is no way to discuss the atomic-resolution details of interfaces/interactions, but the fact that the two domains are in close proximity is obvious according to this peer-reviewed and published in a reputable journal from a reputable lab study.

Second, the structural model shown in Fig. 2g fails to explain why V1397D and I1399D equally disrupted Taf2 interaction with Taf14YEATS and Taf14ET (Fig. 2i). ... I would suggest another possibility that there are two Taf2CT peptides binding to Taf14ET and Taf14YEATS separately. Both copies of Taf2CT form beta-strand to pair with the existing Taf14ET and Taf14YEATS beta-sheets, respectively, and make hydrophobic contacts with Taf14 through Taf2 V1397 and I1399. – again, we would rather avoid overstatements and speculations on the mechanisms of the YEATS-Taf2 complex formation.

Reviewer 1, Comment 2: Association of Taf14 with Taf2 and DNA is essential for transcriptional regulation. In the current manuscript, the authors just showed four genes (their initial submission showed twelve genes) and made conclusions based on these four genes. I think it is cherry-picking. – Fig. 4f was revised in response to previous reviewer’s comments that we need to include statistics and highlight what was statistically different between the constructs and include genes that have been identified as Taf14 targets (ref. 18). Reducing the number of genes in the figure indeed greatly helped to visualize the differences (including all the original genes with statistical analysis results in a very crowded figure which is hard to decipher).

Even for the four genes they showed, two of four genes (ACS1, YKR039W) showed different suppression effects on Taf14 7K/Rmut and Taf14ΔET, so it hardly supported the conclusion “the Taf147K/Rmut linker mutant also showed a loss of Taf14-mediated suppression similar or equal to the loss of Taf14ET” (line 198-199). – this sentence has been revised to: “The Taf14_{7K/Rmut} linker mutant also showed a loss of Taf14-mediated suppression similar or equal to the loss of Taf14_{ET} for YPR145 and YER145.”

In addition, in most circumstances (either in the current four genes or the previous twelve genes), there was not much difference between Taf14 7K/R mutant and W81A mutant, so it is not appropriate to make the statement from lines 198-200. – the phrase ‘in contrast’ has been removed from this sentence.

Moreover, authors claimed that “loss of Taf14 acyl-reading had no impact on the ability of Taf14 to associate with these genes“. This is not true. As shown in Fig4g, the recruitments of Taf14 to two genes (YER145, YKR039W) genes were greatly increased, and the recruitment to ACS1 was mildly increased. This cannot be concluded as “no impact on the ability of Taf14 to associate with these genes“. Why

loss of acylation binding activity could increase the recruitment should be discussed. – this sentence has been revised to: “In contrast, loss of Taf14 acyl-reading greatly increased the ability of Taf14 to associate with *YER145*, *YKR039W* and *ACS1*.”

Besides the inconsistency between experimental data and authors’ conclusions, some data contradict the previous publication from the same lab, but the authors did not discuss why they had different observations for the same set of cells. For example, in their earlier paper (Shanle et al., Gene & Dev 2015), they claimed that Taf14 W81A grew slower than WT in higher (37°C) or lower temperatures (25°C) (Fig. 4d in Gene & Dev 2015). But in the current manuscript, they stated no difference between Taf14 WT and W81A in the growth rate at different temperatures (line 210-211, Figure 4h). – the following sentences have been revised to: “As expected, addition of wild type *TAF14* to *taf14Δ* rescued the observed growth defects, however the rescue effect was diminished in *taf14Δ* cells expressing

Taf14 mutants. Among the mutants, Taf14_{7K/Rmut} rescued the growth defects of the *taf14Δ* cells to the highest degree, while *taf14Δ* cells expressing Taf14_{W81A} grew slower than *taf14Δ* cells expressing wild type Taf14, and loss of the ET domain (Taf14_{ΔET}) led to only a partial rescue of the growth defects, supporting previous findings¹⁹ and pointing to the essential role of this domain in overall multifaceted Taf14 function.”

from G&D, 2015

Of note, there is no contradiction with the previous publication from us, please see Figure on the left: top – data of this study, bottom, data from our previous publication (G&D, 2015). In both cases, Taf14 W81A grows slightly slower.

Reviewer 2, Comment 1: ... Nevertheless, there are still several shortcomings, mostly related to the inconsistency of the results and the lack of appropriate discussion.

1. The results from experiments of Taf14 interactions with H3K9cr-NCP need more thorough explanation.

1. 1. The authors have published previously that the Taf14 YEATS domain interacts specifically with H3K9cr (Andrews et al., 2016; Nat. Chem. Biol. 12:396) and the same result is also presented in this manuscript. However, neither full-length Taf14, nor YEATS domain of it can interact with nucleosomes containing H3K9cr. That discrepancy must be explained and appropriately discussed. – it is well established that histone tails within the nucleosome are less available for the interactions with readers than respective histone peptides.

The following sentence has been added on page 8: Binding of Taf14_{YEATS} to H3K9cr-NCP was also much weaker compared to its binding to H3K9cr peptide¹¹, reflecting the previous findings that histone tails within the nucleosome are not as freely available for the interactions with readers as peptides (Suppl. Fig. 10b)^{16,17}.

1. 2. Some of the presented results are self-contradictory. The authors show that 4K/R mutant is deficient in binding to DNA (Fig. 3C) and to H3K9cr-NCP (Suppl. Fig 10F). On the other hand, addition of Taf2CT to the binding assay activates 4K/R binding to H3K9cr-NCP (Suppl. Fig. 10G). If none of the components (YEATS domain, 4K/R mutant, Taf2CT) can bind DNA or nucleosomes, then how does Taf2CT activate 4K/R binding to H3K9cr-NCP? This has not explained in the manuscript.

There are two DNA-binding motifs in the linker, 4K/R and 3K/R. Mutation of 4K/R decreases but does not eliminate DNA binding (Fig. 3c). In the absence of Taf2, the linker in Taf14(4K/R) is occluded

(Suppl. Fig. 10f), whereas binding of Taf2 releases the linker, so the 3K/R motif binds to the nucleosomal DNA (Suppl. Fig. 10g).

The following sentence has been revised on page 8: However, the Taf2-mediated effect in binding to H3K9cr-NCP was reduced in the case of Taf14_{4K/Rmut}, in which one of the DNA-binding motifs (4K/R) is impaired while another DNA-binding motif (3K/R) remains (Suppl. Fig. 10f, g).

Reviewer 2, Comments 2 and 3: Suppl. Fig. 11. Why is Taf14-ET-delta protein not detected with HA antibody? Apparently, the protein is expressed from the pRS313-Taf14(1-132)-HA3-SSN6 plasmid (Suppl. Table 4) and it should contain a C-terminal 3xHA tag...

Even assuming that Taf14-ET-delta protein is correctly expressed, it lacks the ET domain, which is required for Taf14 interaction with all protein complexes (including TFIID) and the entire linker domain that apparently interacts with DNA...

The labelling of Taf14 constructs throughout the manuscript is confusing...

We apologize for the confusion, information about the Taf14 Δ ET and Taf14(7K/R) constructs used in *in vivo* experiments was missing in Suppl. Table 4. These constructs are now listed in Suppl. Fig. 4. Taf14 Δ ET contains aa 1-174 and does not have the HA tag.

Reviewer 2, Comment 4: The 7K/R mutant of Taf14 appears to be deficient in repression of Taf14-dependent genes (Fig. 4F), while it is fully active in rescuing of taf14-delta cells in the spot test assay (Fig. 4H). Collectively, these results suggest that the DNA binding activity of Taf14 is important for its function (Fig. 4F) and in the same time that it is not important at all (Fig. 4H). This needs a plausible explanation and discussion. – we note that growth on yeast plates and changes at individual genes do not need to correlate. Taf14 is involved in functions of multiple complexes, and it is likely to observe differential effects depending on which and how much Taf14-containing complexes dominate in these distinctive processes.

Line 100: reference to Fig. 1C is not appropriate. – removed

Line 138: reference to Fig. 4C is incorrect, it should be 3C. - corrected

Line 187: reference to Suppl. Fig. 6H,I is incorrect, presumably it should be Suppl. Fig. 10. - corrected

REVIEWER COMMENTS

Reviewer #1 (Remarks to the Author):

Regarding my first comment, authors replied in the current rebuttal letter "this is incorrect, we do not propose or state "that Taf2CT forms a contiguous beta-strand to pair with both Taf14 YEATS and ET beta-sheets" anywhere in the manuscript". I want to explain why I had such an impression. In authors' last rebuttal letter, authors stated that "It is really fascinating and seems the Taf2 peptide pairs with the beta-strand of the YEATS domain in an antiparallel manner. We don't elaborate such a pairing to avoid overstatement. But I am positive that in the cryo-EM structure, a tail of another protein is sandwiched between the YEATS and ET domains (TK).Both beta-strands of the YEATS domain (at one of the edges of the beta sandwich) are perturbed in NMR experiments, suggesting that Taf2 might form the additional beta-strand at this edge (also corroborating the overall orientation of the YEATS-(ligand)-ET assembly in the cryo-EM structure), bridging the existing beta-sheets of the YEATS and ET domains". These sentences gave me the impression that authors tried to establish such a model that Taf2CT beta-pairs with YEATS. The authors have revised the manuscript in addressing my concerns overall. Thus, I support the publication of this manuscript.

Reviewer #2 (Remarks to the Author):

All versions of the manuscript have had one common problem – the authors present their own narrative regardless the actual data. Although this problem has diminished in the course of revisions, it still persists. I can't support the acceptance of this manuscript unless it is dealt thoroughly. These are the examples of deliberate misinterpretations of the data and ignorance of my previous request:

1.

Suppl. Figs. 10a-c show clearly that all the tested proteins behave exactly in the same way in the assay, none of them can bind nucleosomes. However, in the text, the authors present it as a dramatic difference:

"... we found that neither wild type Taf14FL nor the W81A mutant of Taf14FL or Taf14ET appreciably interact with the H3K9cr-containing nucleosome ...", making clear that these proteins do not bind nucleosomes. I agree with that.

However, the next sentence states: " ... **Binding** of Taf14YEATS to H3K9cr-NCP was also much weaker compared to its binding to H3K9cr peptide ...", making clear that Taf14YEATS **binds** nucleosomes, although not as strongly as to the H3K9cr peptide. I don't agree with the statement that it binds nucleosomes. In fact, there is no "much weaker binding", there is "no binding at all".

Ironically, this was appreciated also by the authors in the previous version of the manuscript: "... we found that neither wild type Taf14FL nor the W81A mutant of Taf14FL or **the individual Taf14YEATS** and Taf14ET domains appreciably interact with the H3K9cr-containing nucleosome ...".

2.

Despite the authors did not see binding of any parts of Taf14 to H3K9cr-NCP specifically, they continue to emphasize the importance of this modification in the text:

"... The association with H3K9cr-NCP was detected after the ET domain was removed..."

"... The notable increase in binding to H3K9cr-NCP was also observed when Taf2CT was added..."

"... all the Taf14 constructs containing the linker region, enhanced the formation of the complexes with H3K9cr-NCP..."

"... the Taf2-mediated effect in binding to H3K9cr-NCP was reduced..."

"... The presence of the crotonylated H3K9 modification in the nucleosome, which is recognized by Taf14YEATS, further enhanced the interaction with Taf14..."

Actually, the last sentence is incorrect as the data in the Supplementary Figure 10b show clearly, that YEATS domain does not "recognize" H3K9cr-modified nucleosomes.

The authors should show the experiments with H3K9cr and unmodified nucleosomes side-by-side, if they want to make the point that H3K9cr has a specific role in the nucleosome binding of Taf14 protein. Otherwise it is just an aggressive suggestion of their narrative, without supportive experimental proof. I understand that authors did all experiments with H3K9cr nucleosomes in hope that it would enhance Taf14 binding via its YEAST domain. However, this modification did not discriminate the binding of proteins with functional and non-functional YEATS domains. That is OK and I'm not requesting that all the experiments should be re-done with unmodified nucleosomes. However, after it turned out that H3K9cr does not change anything, its presence should be not emphasized in every second sentence. The current writing style leaves very clear impression that all results here are strictly specific to H3K9cr nucleosomes (which we actually do not know, because H3K9cr and unmodified nucleosomes have not been compared). Quite an opposite, the data in the supplementary figure 10a strongly suggests that H3K9cr has no role whatsoever in these assays.

3.

The authors state:

"The binding of Taf2CT to Taf14FL, Taf14deltaET, Taf14deltaYEATS and Taf14W81A, all the Taf14 constructs containing the linker region, enhanced the formation of the complexes with H3K9cr-NCP". I can't agree in regard to Taf14deltaET (Figs. 4b and S10e) – there is absolutely no effect of Taf2CT in these binding reactions. By the way, it rather suggests that Taf2CT interaction with Taf14YEATS is irrelevant in terms of regulation of Taf14 function.

4.

My previous request to discuss the conflicting results was not appropriately addressed. The authors answered to me, but did not do anything to explain this discrepancy in the manuscript. My request was:

"The 7K/R mutant of Taf14 appears to be deficient in repression of Taf14-dependent genes (Fig. 4F), while it is fully active in rescuing of taf14-delta cells in the spot test assay (Fig. 4H). Collectively, these results suggest that the DNA binding activity of Taf14 is important for its function (Fig. 4F) and in the same time that it is not important at all (Fig. 4H). This needs a plausible explanation and discussion"

We thank the Editor and Reviewer 2 for the insightful and very constructive comments, which were helpful in revising and strengthening this manuscript.

Reviewer 2, Comment 1: ... Suppl. Figs. 10a-c show clearly that all the tested proteins behave exactly in the same way in the assay, none of them can bind nucleosomes...

However, the next sentence states: "... **Binding** of Taf14YEATS to H3K9cr-NCP was also much weaker compared to its binding to H3K9cr peptide ...", making clear that Taf14YEATS **binds** nucleosomes, although not as strongly as to the H3K9cr peptide. I don't agree with the statement that it binds nucleosomes. In fact, there is no "much weaker binding", there is "no binding at all".

– this sentence on page 8 has been revised to: "In contrast to its binding to the H3K9cr peptide (Fig. 2c)¹¹, the individual Taf14_{YEATS} was also incapable of binding to H3K9cr-NCP, reflecting the previous findings that histone tails within the nucleosome are not as freely available for the interactions with readers as peptides (Suppl. Fig. 10b)^{16,17}."

Reviewer 2, Comments 2: Despite the authors did not see binding of any parts of Taf14 to H3K9cr-NCP specifically, they continue to emphasize the importance of this modification in the text:

"... The association with H3K9cr-NCP was detected after the ET domain was removed..."

"... The notable increase in binding to H3K9cr-NCP was also observed when Taf2CT was added..."

"... all the Taf14 constructs containing the linker region, enhanced the formation of the complexes with H3K9cr-NCP..."

"... the Taf2-mediated effect in binding to H3K9cr-NCP was reduced..."

"... The presence of the crotonylated H3K9 modification in the nucleosome, which is recognized by Taf14YEATS, further enhanced the interaction with Taf14..."

– it's not intentional, it is the correct name of this nucleosome. We have replaced the name "H3K9cr-NCP" in the text with just "the nucleosome" in these sentences on page 8. Still, because we also show data with the unmodified NCP (Suppl. Fig. 10h, i), we have to use the correct names.

Actually, the last sentence is incorrect as the data in the Supplementary Figure 10b show clearly, that YEATS domain does not "recognize" H3K9cr-modified nucleosomes.

– this sentence on page 8 has been revised to: "Interestingly, despite the isolated Taf14_{YEATS} did not associate with H3K9cr-NCP (Suppl. Fig. 10b) but bound well to the H3K9cr peptide (Fig. 2c), the interaction of Taf14_{FL} with the unmodified NCP in the presence of Taf2 was enhanced when the crotonylated H3K9 modification was present in the nucleosome (Fig. 4c and Suppl. Fig. 10h, i)." please also see our response below.

The authors should show the experiments with H3K9cr and unmodified nucleosomes side-by-side, if they want to make the point that H3K9cr has a specific role in the nucleosome binding of Taf14 protein. Otherwise it is just an aggressive suggestion of their narrative, without supportive experimental proof.

– the side-by-side comparison of binding of Taf14 to unmodified NCP and H3K9cr-NCP is shown in Suppl. Fig. 10h and Fig. 4c. These data show the enhancement of binding of Taf14 to H3K9cr-NCP in the presence of Taf2 as compared to Taf14 binding to unmodified NCP in the presence of Taf2 (compare lanes 1:1500 in Fig. 4c and Suppl. Fig. 10h).

I understand that authors did all experiments with H3K9cr nucleosomes in hope that it would enhance Taf14 binding via its YEAST domain. However, this modification did not discriminate the binding of proteins with functional and non-functional YEATS domains. That is OK and I'm not requesting that all

the experiments should be re-done with unmodified nucleosomes. However, after it turned out that H3K9cr does not change anything, its presence should be not emphasized in every second sentence. The current writing style leaves very clear impression that all results here are strictly specific to H3K9cr nucleosomes (which we actually do not know, because H3K9cr and unmodified nucleosomes have not been compared).

– we used and compared H3K9cr-NCP and unmodified NCP (Fig. 4c and Suppl. Fig. 10h).

Reviewer 2, Comment 3: The authors state:

“The binding of Taf2CT to Taf14FL, Taf14deltaET, Taf14deltaYEATS and Taf14W81A, all the Taf14 constructs containing the linker region, enhanced the formation of the complexes with H3K9cr-NCP”. I can't agree in regard to Taf14deltaET (Figs. 4b and S10e) – there is absolutely no effect of Taf2CT in these binding reactions...

– this phrase on page 8 has been revised to “The binding of Taf2_{CT} to Taf14_{FL}, Taf14_{ΔYEATS} and Taf14_{W81A...}”

Reviewer 2, Comment 4: My previous request to discuss the conflicting results was not appropriately addressed. The authors answered to me, but did not do anything to explain this discrepancy in the manuscript. My request was:

“The 7K/R mutant of Taf14 appears to be deficient in repression of Taf14-dependent genes (Fig. 4F), while it is fully active in rescuing of taf14-delta cells in the spot test assay (Fig. 4H). Collectively, these results suggest that the DNA binding activity of Taf14 is important for its function (Fig. 4F) and in the same time that it is not important at all (Fig. 4H). This needs a plausible explanation and discussion”

- The following sentences have been added on page 10: “While the DNA-binding activity of Taf14 was largely dispensable for the yeast growth in the spot assays (Fig. 4h), the Taf14_{7K/Rmut} was deficient in repression of Taf14-mediated suppression of a set of genes (Fig. 4f). The observed differential effects indicate that distinctive functions or a different degree of these functions of Taf14 are necessary for these processes. As Taf14 is a component of multiple complexes, differential effects can also arise depending on which and to what extent the Taf14-containing complexes dominate in these processes.”